# PPAR-γ regulates the effector function of human T helper 9 cells by promoting glycolysis

Nicole L. Bertschi [1,6], Oliver Steck[1,6], Fabian Luther[1,6], Cecilia Bazzini[1], Leonhard von Meyenn [1], Stefanie Schärli [1], Angela Vallone[1], Andrea Felser[2], Irene Keller[3], Olivier Friedli [4], Stefan Freigang [4], Nadja Begré[1], Susanne Radonjic-Hoesli [1], Cristina Lamos[1], Max Philip Gabutti[1], Michael Benzaquen [1], Markus Laimer [5], Dagmar Simon[1], Jean-Marc Nuoffer [2] & Christoph Schlapbach [1] ✉

T helper 9 (T$_H$9) cells promote allergic tissue inflammation and express the type 2 cytokines, IL-9 and IL-13, as well as the transcription factor, PPAR-γ. However, the functional role of PPAR-γ in human T$_H$9 cells remains unknown. Here, we demonstrate that PPAR-γ drives activation-induced glycolysis, which, in turn, promotes the expression of IL-9, but not IL-13, in an mTORC1-dependent manner. In vitro and ex vivo experiments show that the PPAR-γ-mTORC1-IL-9 pathway is active in T$_H$9 cells in human skin inflammation. Additionally, we find dynamic regulation of tissue glucose levels in acute allergic skin inflammation, suggesting that in situ glucose availability is linked to distinct immunological functions in vivo. Furthermore, paracrine IL-9 induces expression of the lactate transporter, MCT1, in T$_H$ cells and promotes their aerobic glycolysis and proliferative capacity. Altogether, our findings uncover a hitherto unknown relationship between PPAR-γ-dependent glucose metabolism and pathogenic effector functions in human T$_H$9 cells.

T helper (T$_H$) cells have evolved into distinct subsets that mediate specific immune responses, protecting the host from various infectious and noninfectious challenges[1]. However, impaired T$_H$ cell function can lead to inflammatory disease. Emerging evidence from both mice and humans indicates that type 2-driven diseases are mediated by a distinct subpopulation of T$_H$2 cells, referred to as pathogenic T$_H$2 (pT$_H$2) cells[2,3]. Thus, pT$_H$2 cells and their effector molecules serve as prime targets for novel therapeutic approaches. In fact, pT$_H$2 cells from a wide range of diseases such as allergic asthma, eosinophilic esophagitis (EoE), nasal polyps, and allergic contact dermatitis (ACD) share a common underlying transcriptome and overlapping functional characteristics[4–10]. In particular, pT$_H$2 cells express the peroxisome proliferator-activated receptor gamma (PPAR-γ) transcription factor and the IL-17RB and IL-9R cytokine receptors. They also secrete high levels of interleukin (IL-)13, IL-5, and IL-9[2]. Interestingly, PPAR-γ antagonism in human T$_H$ cells has been shown to inhibit the production of IL-9, while mice with T$_H$ cell-specific *Pparg* knockout are protected against T$_H$2-mediated immunopathology[11,12]. This suggests that PPAR-γ plays an important functional role in pT$_H$2 cells. Although PPAR-γ is intricately linked to the pT$_H$2 cell phenotype, the mechanisms by which PPAR-γ regulates pT$_H$2 cell function remain largely unknown.

[1]Department of Dermatology, Inselspital, Bern University Hospital, University of Bern, Bern, Switzerland. [2]Institute of Clinical Chemistry, University of Bern, Bern, Switzerland. [3]Interfaculty Bioinformatics Unit and Swiss Institute of Bioinformatics, University of Bern, Bern, Switzerland. [4]Institute of Tissue Medicine and Pathology, University of Bern, Bern, Switzerland. [5]Department of Diabetes, Endocrinology, Nutritional Medicine and Metabolism (UDEM), Bern University Hospital, University of Bern, Bern, Switzerland. [6]These authors contributed equally: Nicole L. Bertschi, Oliver Steck, Fabian Luther. ✉ e-mail: christoph.schlapbach@insel.ch

The functional investigation of PPAR-γ in human $T_H$ cells has been hampered by the low frequency of PPAR-γ-expressing $T_H$ cells in human peripheral blood. However, we have recently identified IL-9-producing $T_H$9 cells as a subpopulation of PPAR-γ$^+$ $T_H$2 cells that possesses key characteristics of p$T_H$2 cells[9]: Human $T_H$9 cells reside within the CCR4$^+$/CCR8$^+$ population of effector memory T cells ($T_{EM}$). Functionally, they produce high levels of IL-9 and IL-13 and express transcription factors associated with the p$T_H$2 lineage. Both in vitro and in vivo primed $T_H$9 cells are distinct from conventional $T_H$2 (c$T_H$2) cells: $T_H$9 cells produce IL-9 in a transient activation-dependent manner and express PPAR-γ, which they rely on for their full effector function. Due to these key similarities shared with p$T_H$2 cells, $T_H$9 cells represent a valuable tool for studying the functional role of PPAR-γ in human $T_H$2 cell biology.

PPAR-γ is a ligand-activated nuclear receptor classically known for regulating lipid and glucose metabolism in adipocytes and other mesenchymal cells[13]. PPAR-γ is activated by synthetic ligands, such as thiazolidinediones, as well as endogenous ligands thought to be derived from fatty acids[14,15]. Once activated, PPAR-γ dimerizes with the retinoid X receptor (RXR) and binds to genomic PPAR-responsive regulatory elements (PPREs) to control the expression of genes involved in lipid and glucose metabolism, as well as inflammation[13]. The functional role of PPAR-γ in $T_H$2 cells is best described in murine models, where it promotes IL-33R expression and thereby enhances the sensitivity of $T_H$2 cells to tissue alarmins in allergic inflammation. Accordingly, mice with CD4$^+$ T cell-specific *Pparg* knockout exhibit impaired antiparasitic immunity, which protects them from allergic lung inflammation[11,12]. In humans, however, the function of PPAR-γ in p$T_H$2 cells and its role in allergy are yet to be characterized.

Here, we sought to investigate the mechanism by which PPAR-γ regulates the effector function of human $T_H$9 cells, which share key characteristics with p$T_H$2 cells. Collectively, our data point to the central role of PPAR-γ in promoting aerobic glycolysis, activating mTORC1, and stimulating IL-9 production in $T_H$9 cells. Further, we uncover a previously unknown functional role of paracrine IL-9 in promoting metabolic adaptation to high-glucose environments in acute allergic skin inflammation. Accordingly, PPAR-γ, IL-9, and their downstream targets might represent therapeutic leverage points in ACD and type 2-driven diseases.

## Results

### In vitro and in vivo primed $T_H$9 cells display key features of pathogenic $T_H$2 cells

The transcriptomic signature of p$T_H$2 cells has been previously identified by single-cell analysis of T cells extracted from multiple $T_H$2-driven diseases[2]. To test whether human in vitro primed $T_H$9 cells recapitulate the core p$T_H$2 cell phenotype, we differentiated naive T cells into $T_H$1 (IL-12), $T_H$2 (IL-4), $T_H$9 (IL-4+TGF-β), and i$T_{REG}$ (TGF-β) cells and performed transcriptomic profiling using RNA sequencing (RNA-seq) at day 7. Pairwise comparison to other subsets showed that 1492 genes were specifically upregulated in $T_H$9 cells (Fig. 1a). We then compared our $T_H$9 transcriptome with three p$T_H$2-specific transcriptomes identified in EoE[4], allergic asthma[5], and allergen-specific $T_H$2 cells[7], respectively (Fig. 1b). *PPARG, IL5, IL17RB*, and *IL9R*, which are hallmarks of p$T_H$2 cells, were upregulated in $T_H$9 cells as well as in all three p$T_H$2 datasets, while *IL9* was upregulated in two of the three (Fig. 1c and Supplementary Fig. 1a). In contrast, *SPI1*, encoding the transcription factor PU.1, previously shown to be associated with IL-9 expression[16,17], was neither expressed in p$T_H$2-specific transcriptomes, nor in $T_H$9 cells (Supplementary Fig. 1a). High levels of PPAR-γ were confirmed at the protein level in both in vitro primed $T_H$9 cells and in vivo primed $T_H$9 clones, which we generated from ex vivo isolated memory $T_H$ cells that were sorted based on chemokine receptor expression (Fig. 1d–f). In summary, these findings strongly support our hypothesis that in vitro and in vivo primed $T_H$9 cells share key similarities with p$T_H$2 cells.

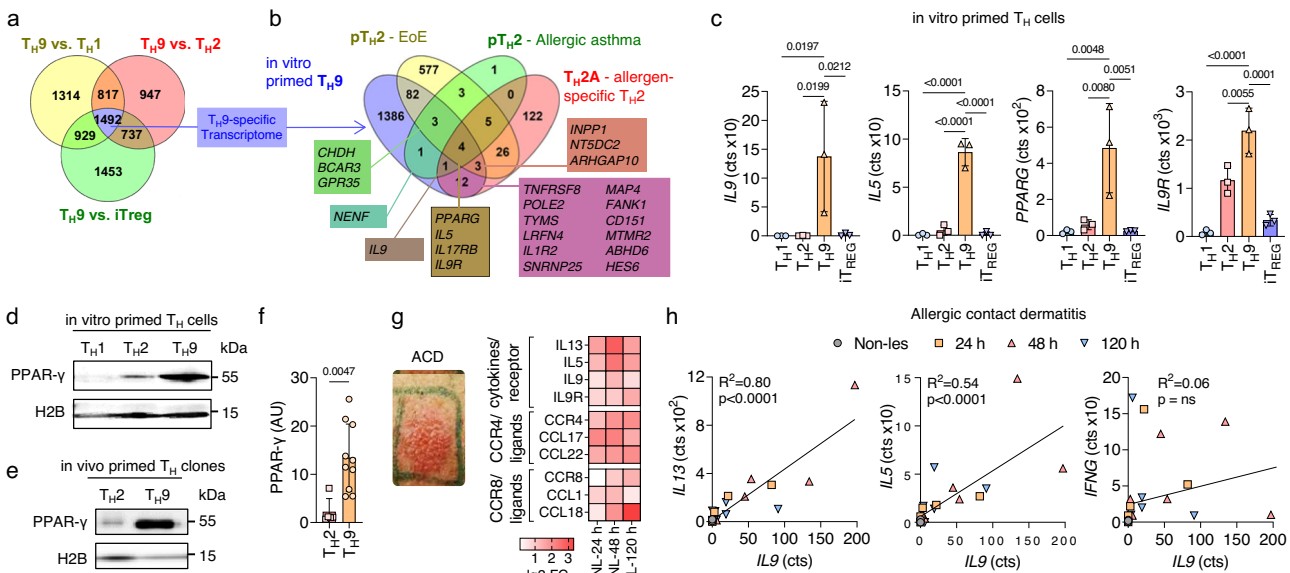

**Fig. 1 | In vitro and in vivo primed $T_H$9 cells display key features of pathogenic $T_H$2 cells. a** Venn diagram of RNA-seq data from $T_H$ cell subsets primed in vitro showing the number of genes significantly upregulated in $T_H$9 cells compared to other cell subsets ($p_{adj}$ <0.05). **b** Venn diagram of $T_H$9-specific transcriptome identified in **a** and p$T_H$2-associated genes identified in eosinophilic esophagitis (EoE)[4], allergic asthma[5], and allergen-specific $T_H$2 cells ($T_H$2A)[7]. **c** Expression of selected p$T_H$2-associated genes as determined in **a**. **d**–**f** Western blot analysis of PPAR-γ in different $T_H$ cell subsets primed in vitro (**d**) and in $T_H$2 and $T_H$9 clones primed in vivo (**e**, **f**). **g** Changes in gene expression of selected p$T_H$2-associated genes. **h** In-sample correlations of T cell cytokines with *IL9*. The data are representative of independent experiments with three (**a**–**c**) or six (**g**, **h**) donors or five (**f** ($T_H$2)) or 10 (**f** ($T_H$9)) clones from two donors. Statistics: **a** differences between cell subsets were calculated as an adjusted log-fold change, and hypothesis testing was performed using the Benjamini–Hochberg adjusted *p* value (DESeq2). **c** One-way ANOVA, followed by a Dunnett's test for multiple comparisons. **f** Two-tailed unpaired *t* test. **h** Simple linear regression. The data are presented as mean ± SD. Only *p* values <0.05 are shown.

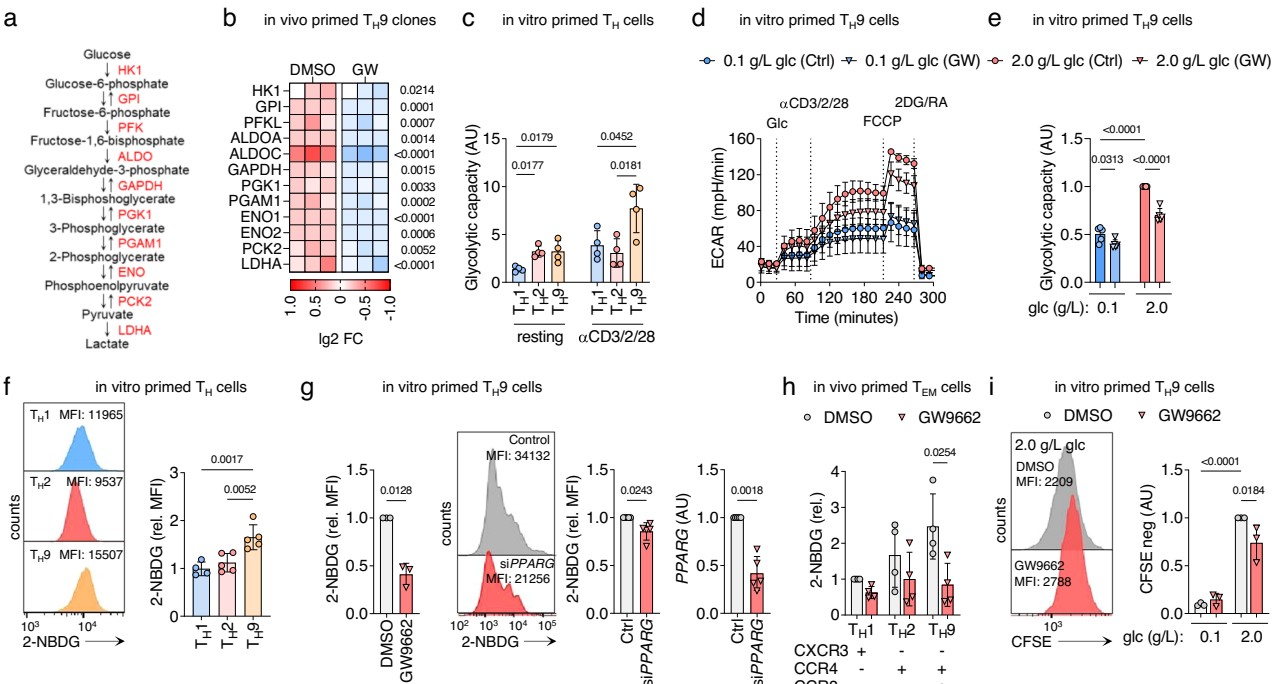

**Fig. 2 | PPAR-γ mediates the high glycolytic activity of $T_H9$ cells. a** Intermediates and enzymes (red) of aerobic glycolysis. **b** RNA-seq of $T_H9$ clones incubated in presence of GW9662 for 48 h and activated by αCD3/CD2/CD28 for 12 h. **c** Maximal glycolytic capacity of in vitro primed $T_H$ cells in the resting state and 24 h after activation with αCD3/CD2/CD28. **d** ECAR measurements of in vitro primed $T_H9$ cells cultured in media with glucose of different levels and GW9662 for 48 h and activated by injection of glucose and αCD3/CD2/CD28. **e** Maximal glycolytic capacity of in vitro primed $T_H9$ cells from **d**. **f** Glucose uptake by in vitro primed $T_H$ cells measured with fluorescent 2-NBDG uptake by flow cytometry at day 7. **g** Glucose uptake by naive $T_H$ cells primed under $T_H9$ conditions for 7 days in presence of GW9662 or transfected with *PPARG* and control siRNA, respectively. Efficiency of knockdown was determined by measuring *PPARG* levels after transfection by RT-qPCR (right). **h** Glucose uptake of in vivo primed effector memory $T_H$ cells ($T_{EM}$) sorted by flow cytometry into $T_H1$, $T_H2$, and $T_H9$ cells according to their chemokine receptor profile. Sorted $T_H$ cells were incubated in presence or absence of GW9662 for 48 h, and activated by αCD3/CD2/CD28 for 4 h. **i** Proliferation of in vitro primed $T_H9$ cells, activated by αCD3/CD2/CD28 for 4 days in presence or absence of GW9662 and glucose of different levels, measured with CFSE dilution by flow cytometry. The data are representative of one experiment with three clones from one donor (**b**) or independent experiments with three (**d**, **g** (left), **i**), four (**c**, **f** ($T_H1$), **h**), five (**e**, **f** ($T_H2$ and $T_H9$), **g** (right)) or six (**g**, right) donors. Statistics: **b** differences between treatment groups were calculated as an adjusted log-fold change, and hypothesis testing was performed using the Benjamini–Hochberg adjusted $p$ value (DESeq2). **c**, **f**, **h** One-way ANOVA, followed by a Tukey's test for multiple comparisons. **g** Two-tailed paired $t$ test. **e**, **i** One-way ANOVA, followed by a Šidák's test for multiple comparisons. The data are presented as mean ± SD. Only $p$ values <0.05 are shown.

As we had previously identified PPAR-γ⁺ $T_H9$ cells in ACD[9], we next considered validating the association between the $T_H9$ and $pT_H2$ phenotypes in acute allergic skin inflammation. Time course transcriptomics of untreated non-lesional (NL) skin and positive patch test reactions of lesional skin to nickel at 24 h, 48 h, and 120 h post allergen application showed a marked upregulation of the $pT_H2$-associated genes in ACD (Fig. 1g and Supplementary Fig. 1b). Across individual samples, the expression of *IL9* correlated with the expression of *IL13* ($R^2 = 0.80$; $P < 0.0001$), *IL5* ($R^2 = 0.54$; $P < 0.0001$), *IL31* and *IL19*, but not *IFNG* (Fig. 1h and Supplementary Fig. 1c), consistent with the predominance of $T_H9$ cells in the $T_H2$ cell pool. Due to the fact that *PPARG* is expressed in various skin cell types, including keratinocytes[9], the correlation analysis of *PPARG* with *IL9* does not allow any conclusion with regard to $T_H9$ cells.

Collectively, these data show that both in vitro and in vivo primed $T_H9$ cells express the core features of $pT_H2$ cells, including upregulated expression of PPAR-γ, IL-5, IL-9, and IL-9R. They can thus serve as model cells for studying the functional role of PPAR-γ in human $T_H$ cells. Further, human ACD appears to be a valid model for studying the functionality of $T_H9$ cells ex vivo.

## PPAR-γ mediates the high glycolytic activity of $T_H9$ cells

To investigate the role of PPAR-γ in human $T_H$ cells, we first assessed the transcriptional response of activated $T_H9$ clones to treatment with GW9662, a potent PPAR-γ antagonist. Pathway analysis of RNA-seq data revealed concerted downregulation of genes associated with T cell activation, glucose metabolism, and aerobic glycolysis (Supplementary Fig. 2a). At the single gene level, mRNA expression of all aerobic glycolysis enzymes was significantly downregulated by PPAR-γ inhibition (Fig. 2a, b). This prompted us to further analyze the role of PPAR-γ in aerobic glycolysis of $T_H9$ cells. In vitro primed $T_H9$ cells showed higher glycolytic activity than $T_H1$- or $T_H2$-primed T cells after activation with αCD3/CD2/CD28 (Fig. 2c and Supplementary Fig. 2b). To verify whether PPAR-γ was involved in this process, we starved in vitro primed $T_H9$ cells in glucose-free medium in presence or absence of GW9662. We then performed measurements of oxygen consumption rate (OCR) and extracellular acidification rate (ECAR) in real-time before and after activation with αCD3/CD2/CD28 in either low or high-glucose environments. PPAR-γ inhibition by GW9662 or the alternative PPAR-γ antagonist, T0070907, hampered the glycolytic response in high- but not low-glucose environments, particularly following T cell activation (Fig. 2d, e and Supplementary Fig. 2c, d). These findings were corroborated by measurements of glucose uptake, in which in vitro primed $T_H9$ cells showed higher glucose uptake compared to $T_H1$, and $T_H2$ cells (Fig. 2f). Importantly, glucose uptake in $T_H9$ cells was reduced by PPAR-γ-inhibition or by siRNA-induced *PPARG* knockdown (Fig. 2g).

We next expanded our findings to in vivo primed memory $T_H$ cells, leveraging the ability to sort different subsets ex vivo based on their chemokine receptor profile, including CXCR3⁻/CCR4⁺/CCR6⁻/CCR8⁺

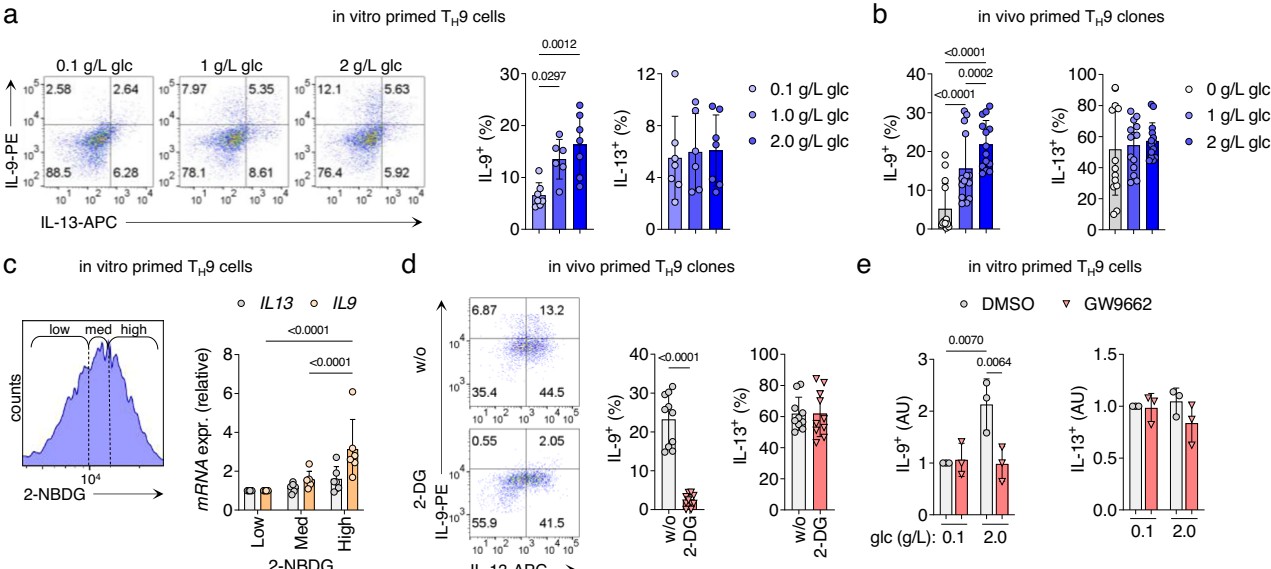

**Fig. 3 | High glycolytic activity of T$_H$9 cells regulates specific effector functions.**
**a** Cytokine expression measured by flow cytometry of T$_H$9 cells primed in vitro in media containing glucose of different levels for 7 days. **b** Cytokine expression of in vivo primed T$_H$9 clones cultured for 72 h in media containing glucose of different levels, measured as in **a**. **c** In vitro primed T$_H$9 cells were activated for 4 h with αCD3/CD2/CD28, then sorted by flow cytometry based on their glucose uptake measured by 2-NBDG uptake (left). Cytokine expression in the sorted T$_H$ cell populations was measured by RT-qPCR (right). **d** Cytokine expression of in vivo primed T$_H$9 cells cultured for 7 days in the presence of 2-DG, measured as in **a**.

**e** Cytokine expression of in vitro primed T$_H$9 cells cultured for 48 h in media containing glucose of different levels and in presence or absence of GW9662, measured as in **a**. The data are representative of independent experiments with three (**e**), six (**c**), or seven (**a**) donors or fourteen clones from two donors (**b**, **d**). Statistics: **a**, **c** One-way ANOVA, followed by a Tukey's test for multiple comparisons. **b**, **e** One-way ANOVA, followed by a Šidák's test for multiple comparisons. **d** Two-tailed paired *t* test. The data are presented as mean ± SD. Only *p* values <0.05 are shown.

T$_H$9 cells[9,18]. Consistent with our previous observations, in vivo primed T$_H$9 cells showed higher glucose uptake compared to other T$_H$ cell subsets, and GW9662 significantly inhibited the glycolytic activity in T$_H$9 cells but not in T$_H$2 or T$_H$1 cells (Fig. 2h).

Given the central role of aerobic glycolysis for T cell proliferation[19], we finally tested whether PPAR-γ inhibition affected TCR stimulation-induced proliferation in low and high-glucose environments. Both GW9662 and T0070907 significantly reduced αCD3/CD2/CD28-induced proliferation in high-glucose environments (Fig. 2i and Supplementary Fig. 2e, f).

Since PPAR-γ has been implicated in mediating fatty acid (FA) uptake in activated T$_H$ cells[20,21], we examined FA metabolism in response to PPAR-γ antagonism. In vitro and in vivo primed T$_H$9 cells did not exhibit higher FA uptake compared with T$_H$1 and T$_H$2 cells (Supplementary Fig. 2g) and PPAR-γ antagonism had no effect on FA uptake (Supplementary Fig. 2h, i). In addition, glutamine uptake of T$_H$9 was not affected by PPAR-γ inhibition, suggesting that glutaminolysis is not primarily regulated by PPAR-γ in these cells (Supplementary Fig. 2j).

PPAR-γ has previously been shown to regulate the expression of IL-33R in murine T$_H$2 cells, thereby increasing their sensitivity to the tissue alarmin IL-33[11,12] in allergic inflammation. While IL-33R (*IL1RL1*) is also associated with the pT$_H$2 phenotype in humans[4,6-8,22], PPAR-γ antagonism did not downregulate its expression in T$_H$9 cells (Supplementary Fig. 2k).

Collectively, these data strongly suggest that both in vitro and in vivo primed human T$_H$9 cells are characterized by high glycolytic capacity post-TCR-stimulation. Moreover, in the setting studied here, PPAR-γ signaling appears to be a crucial mediator of glycolysis, whereas FA oxidation and glutaminolysis remain unaffected.

## High glycolytic activity in T$_H$9 cells differentially regulates cytokine expression

PPAR-γ is required for the full effector function in T$_H$9 and pT$_H$2 cells, including the production of proinflammatory cytokines[9,11,12]. Based on our findings, we hypothesized that PPAR-γ might control cytokine production indirectly, namely by promoting glycolysis. To test this, we cultured in vitro primed T$_H$9 cells in media containing different glucose concentrations and measured their cytokine profiles at day 7. Production of the pT$_H$2 marker cytokine IL-9 showed a strong dependency on glucose availability, whereas production of IL-13, the conventional T$_H$2 cytokine, did not (Fig. 3a). Further, in vivo primed T$_H$9 clones cultured in different glucose concentrations downregulated the production of IL-9 but not IL-13 in low-glucose environments (Fig. 3b).

To demonstrate a direct relationship between high-glucose metabolism and cytokine production, we next sorted in vitro primed T$_H$9 cells according to their glucose uptake level, measured by the uptake of 2-NBDG. Subsequently, we performed RT-qPCR for *IL9*, *IL13*, and *PPARG*. Glucose uptake correlated with the expression of *IL9* and *PPARG*, but not *IL13* (Fig. 3c and Supplementary Fig. 3c). Taken together, these data strongly suggests that glycolytic activity regulates the expression of IL-9. Similar regulation, albeit less pronounced, was observed for *IL5* expression (Supplementary Fig. 3a–c).

In a next step, we hence inhibited glycolysis in T$_H$9 cells using the glucose analog 2-deoxy-d-glucose (2-DG) and the aerobic glycolysis inhibitor lonidamine (LND) to investigate the effect on cytokine expression. In T$_H$9 cells primed in vivo, 2-DG and LND inhibited the expression of IL-9 and IL-5 but not IL-13 (Fig. 3d and Supplementary Fig. 3d, e). Finally, PPAR-γ antagonism in high-glucose environments reduced the production of IL-9 to the levels seen in low-glucose environments, whereas IL-13 levels remained unaffected neither by PPAR-γ antagonism nor by low-glucose availability (Fig. 3e).

To investigate the contribution of FA metabolism to the regulation of cytokines in T$_H$9 cells, we analyzed IL-9 expression in response to FA inhibition. Neither did culturing of T$_H$9 clones in FA-free medium affect IL-9 or IL-13 expression (Supplementary Fig. 3f), nor did inhibition of FA metabolism with etomoxir, an inhibitor of carnitine palmitoyltransferase-1 (Supplementary Fig. 3g).

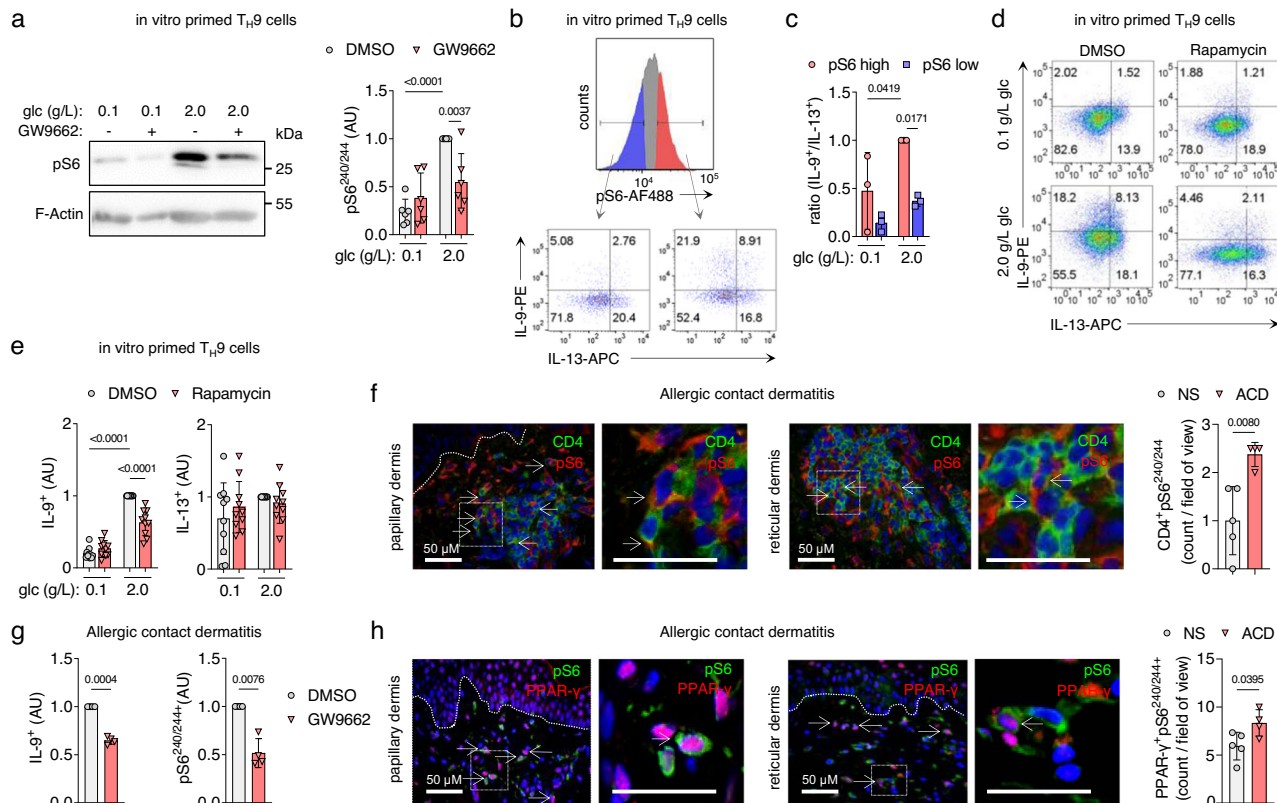

**Fig. 4 | mTORC1 integrates bioenergetics with effector function in $T_H9$ cells.**
**a** Western blot analysis of pS6 in $T_H9$ cells primed in vitro. **b** In vitro primed $T_H9$ cells were cultured in glucose of different levels for 48 h, and pS6 and IL-9 were measured 18 h after activation with αCD3/CD2/CD28 by flow cytometry. The histogram shows pS6 positive cells, split into high and low pS6. Dot-plots represent IL-9+/IL-13+ clusters. **c** The IL-9+/IL-13+ ratio in $T_H9$ cells primed in vitro from **b**. **d**, **e** Cytokine expression of in vitro primed $T_H9$ cells after incubation in glucose of different levels and rapamycin for 48 h, measured as in **b**. **f** Immunofluorescence staining for CD4 and pS6 on skin samples of allergic contact dermatitis (ACD) and quantification of CD4+pS6+ cells in normal skin (NS) and ACD skin samples. Scale

bars, 50 µM. **g** Cytokine expression of T cells isolated from ACD skin biopsies incubated with GW9662 for 48 h, measured as in **b**. **h** Immunofluorescence staining for pS6 and PPAR-γ on skin samples of ACD and quantification of PPAR-γ+pS6+ cells in NS and ACD skin samples. Scale bars, 50 µM. The data are representative of independent experiments with one (**g**), three (**c**), four (**f** (ACD), **h** (ACD)), five (**f** (NS), **h** (NS)), six (**a**), or nine (**e**) donors. Statistics: **a**, **c**, **e** One-way ANOVA, followed by a Šidák's test for multiple comparisons. **f**, **h** Two-tailed unpaired *t* test. **g** Two-tailed paired *t* test. The data are presented as mean ± SD. Only *p* values <0.05 are shown.

Taken together, these observations indicate a dichotomous role of glycolytic activity in regulating the production of IL-9, IL-5, and IL-13 by activated $T_H9$ cells.

## mTORC1 integrates glycolytic activity with the effector function in $T_H9$ cells

We next investigated the mechanisms underlying the association between glycolysis and cytokine production in activated $T_H9$ cells. Mammalian target of rapamycin complex 1 (mTORC1) is a central regulator of cellular metabolism and effector functions in T cells. Nutrients, such as glucose, are critical activators of mTORC1[23]. Furthermore, the mTORC1-hypoxia-inducible factor-1α (HIF-1α) pathway is necessary for the expression of IL-9 in murine T cells, with HIF-1α binding directly to the *Il9* promoter and activating its transcription[24–26]. Given the role of the established mTORC1-HIF-1α-IL-9 axis and our previous results, we hypothesized that mTORC1 might mediate the PPAR-γ-dependent expression of IL-9 in $T_H9$ cells.

Phosphorylation of mTORC1 in activated $T_H9$ cells measured by phosphorylated S6 (pS6) was glucose-dependent and reduced by PPAR-γ inhibition under high-glucose conditions (Fig. 4a, and Supplementary Fig. 4a, b). Indeed, IL-9+ T cells were strongly enriched in the pS6+ cell population, whereas the proportion of IL-13+ T cells was similar in pS6− and pS6+ populations (Fig. 4b, c). Moreover, inhibition of mTORC1 by either siRNA against *RPTOR* or by rapamycin decreased

the production of IL-9 but not IL-13 in activated $T_H9$ cells (Fig. 4d, e and Supplementary Fig. 4c, d).

Since the PPAR-γ agonist troglitazone (TGZ) unexpectedly reduced IL-9 expression in $T_H9$ cells, we next investigated the mechanism by which this occurs. Interestingly, pS6 levels were reduced in the presence of TGZ (Supplementary Fig. 4e), suggesting that mTORC1 is inhibited. Previous studies have shown that PPAR-γ agonists, such as TGZ, activate AMP-activated protein kinase (AMPK)[27]. We, therefore, hypothesized that TGZ-mediated AMPK activation negatively regulates mTORC1, which in turn suppresses IL-9 expression. Indeed, Western blot analysis revealed that TGZ leads to phosphorylation of AMPK and mTORC1 inhibition (Supplementary Fig. 4f). In addition, the AMPK activator A-769662 also reduced IL-9, but not IL-13 levels (Supplementary Fig. 4g). Together, this data strongly supports our hypothesis that IL-9 expression is mTORC1-dependent.

To verify whether $T_H9$ cells expressed activated mTORC1 in human skin inflammation, we performed immunofluorescence staining of normal skin (NS) and ACD skin samples and isolated T cells from such lesions. Double immunofluorescence revealed that CD3+ and CD4+ $T_H$ cells that express pS6 are significantly enriched in the infiltrate of ACD compared to NS (Fig. 4f and Supplementary Fig. 4h, i). Virtually all IL-9+ $T_H$ cells isolated from ACD show S6 phosphorylation, and thus have active mTORC1 signaling (Supplementary Fig. 4j). Incubation with GW9662 ex vivo showed reduced activation of mTORC1 and a significantly inhibited IL-9 production (Fig. 4g). As it is known that ACD

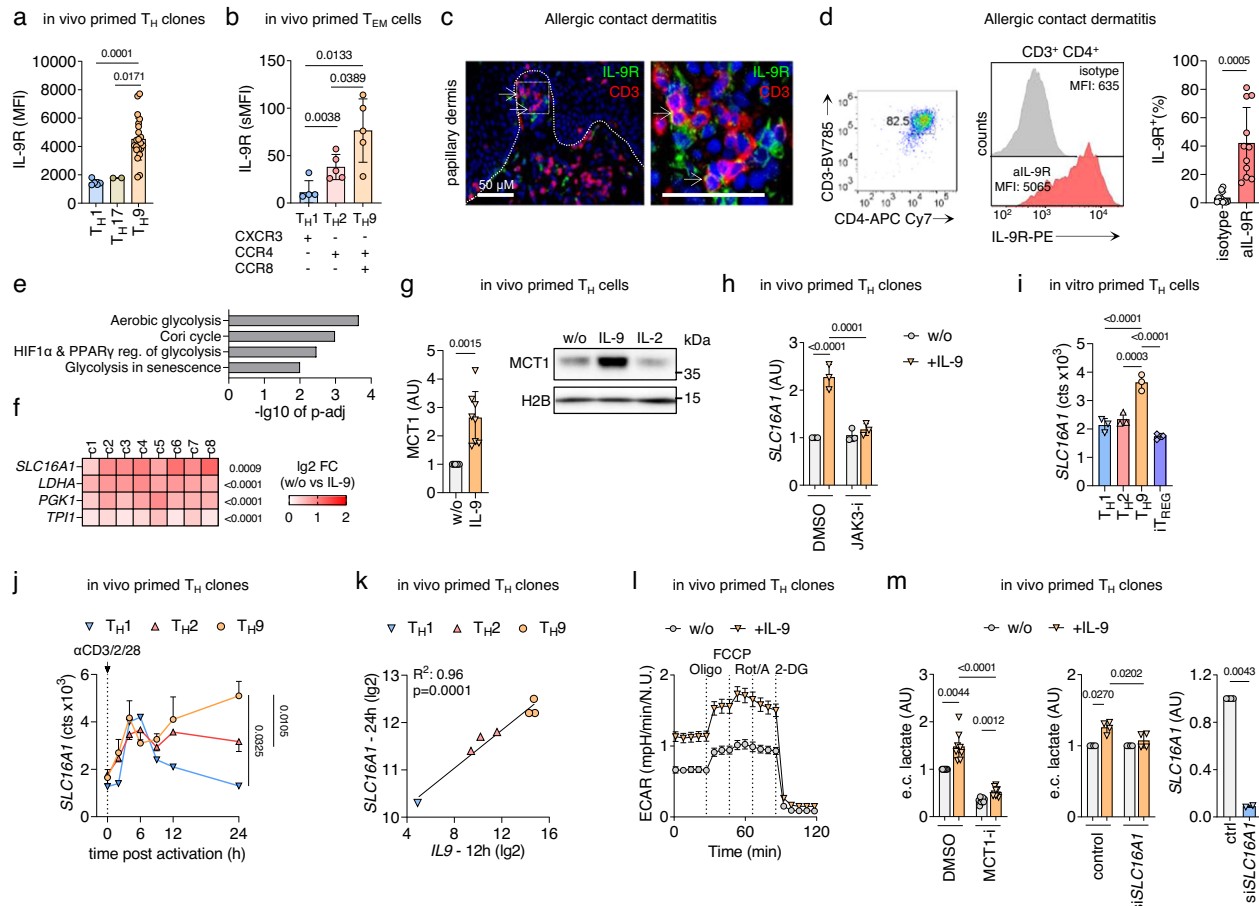

**Fig. 5 | Paracrine IL-9 promotes aerobic glycolysis in IL-9R⁺ $T_H$ cells by inducing the lactate transporter MCT1. a** IL-9R levels of in vivo primed $T_H$ clones analyzed by flow cytometry. **b** IL-9R levels of PBMCs stained for their chemokine receptor profiles analyzed by flow cytometry. **c** Immunofluorescence staining for CD3 and IL-9R on ACD skin. Scale bars, 50 μM. **d** IL-9R levels of T cells isolated from ACD analyzed by flow cytometry. **e–f** RNA-seq of IL-9R⁺ $T_H$ cells isolated from blood and ACD in presence of IL-9 shows (**e**) pathway analysis of the 250 most significant IL-9-induced genes and (**f**) changes in the expression of selected aerobic glycolysis genes. **g** Western blot analysis of MCT1 expression in IL-9R⁺ $T_H$ clones incubated with IL-9 or IL-2 for 48 h. **h** *SLC16A1* expression measured by RT-qPCR in IL-9R⁺ $T_H$ clones in presence of JAK3 inhibitor (JAK3-i) ritlecitinib and IL-9 for 24 h. **i** RNA expression of *SLC16A1* in in vitro primed $T_H$ cells after 7 days. **j, k** Time course transcriptomic data[9] shows RNA expression levels of *SLC16A1* in **i** and correlation between *IL9* and *SLC16A1* expression in **j**. **l** ECAR measurements of in vivo primed IL-9R⁺ $T_H$ clones incubated with IL-9 for 16 h. **m** IL-9R⁺ $T_H$ clones incubated with the

MCT1 inhibitor (MCT1-i) BAY-8002 or transfected with *SLC16A1* siRNA. Extracellular (e.c.) lactate was measured with the Lactate-Glo™ Assay (Promega) after 48 h in presence of IL-9. The data are representative of one experiment with one (**c, l**) donor or two (**a** ($T_H$17)), three (**h**), five (**a** ($T_H$1)) or twenty-two (**a** ($T_H$9)) clones from one donor or independent experiments with eight clones from one (**g**) or two (**e, f**) donors or nine clones from two donors (**m** (left)) or two (**m** (right)), three (**i**), five (**b**) six (**j, k**) or eleven (**d**) donors. Statistics: **a** Two-tailed unpaired *t* test. **b, d, g, m** (right) Two-tailed paired *t* test. **e** Fisher's one-tailed test. **f** Differences between treatment groups were calculated as an adjusted log-fold change, and hypothesis testing was performed using the Benjamini–Hochberg adjusted *p* value (DESeq2). **h, l** One-way ANOVA, followed by a Dunnett's test for multiple comparisons. **k** Simple linear regression. **h, j, l, m** One-way ANOVA, followed by a Tukey's test for multiple comparisons. The data are presented as mean ± SD. Only *p* values <0.05 are shown.

skin is infiltrated by a substantial number of PPAR-γ⁺ $T_H$ cells[9], we finally investigated whether those cells would show the activation of mTORC1. Indeed, PPAR-γ⁺pS6⁺ double-positive lymphocytes were significantly increased in the dermis of ACD compared to NS (Fig. 4h and Supplementary Fig. 4k).

Collectively, these findings strongly suggest that glucose- and PPAR-γ-dependent production of IL-9 in $T_H$9 cells is regulated via mTORC1 in acute allergic skin inflammation.

### Paracrine IL-9 promotes aerobic glycolysis in IL-9R⁺ $T_H$ cells by inducing the lactate transporter MCT1

After revealing the association between PPAR-γ-dependent glycolytic activity and IL-9 production in $T_H$9 cells, we hypothesized that paracrine IL-9 might regulate glucose metabolism and downstream effector functions in IL-9R⁺ $T_H$ cells. Previous data suggested that *IL9R* is preferentially expressed in p$T_H$2 and $T_H$9 cells (Fig. 1 and ref. 4,5,7,8), but these findings had to be confirmed at the protein

level and in human skin inflammation. Thus, we first confirmed the expression of IL-9R in $T_H$9 clones primed in vivo (Fig. 5a), circulating CXCR3⁻/CCR4⁺/CCR8⁺ effector memory $T_H$ cells (Fig. 5b), and $T_H$ cells infiltrating lesions of ACD (Fig. 5c, d), showing that p$T_H$2 and $T_H$9 cells are important targets of IL-9 in human skin inflammation. Next, we performed RNA-seq of IL-9R⁺ $T_H$ clones and IL-9R⁺ $T_H$ cells isolated from ACD skin biopsies, incubated with or without IL-9. The pathway analysis of the 250 most IL-9-induced genes showed a coordinated induction of genes involved in aerobic glycolysis (Fig. 5e), most prominently *SLC16A1* (Fig. 5f). *SLC16A1* encodes the monocarboxylate transporter 1 (MCT1), enabling the rapid export of lactate across the plasma membrane and thereby exerting a glycolytic flux-controlling function[28]. IL-9-induced expression of MCT1 in $T_H$9 clones was confirmed at the protein level (Fig. 5g). In contrast, inhibition of JAK3, which is central to IL-9R signal transduction[29], by ritlecitinib suppressed IL-9-induced upregulation of *SLC16A1* (Fig. 5h).

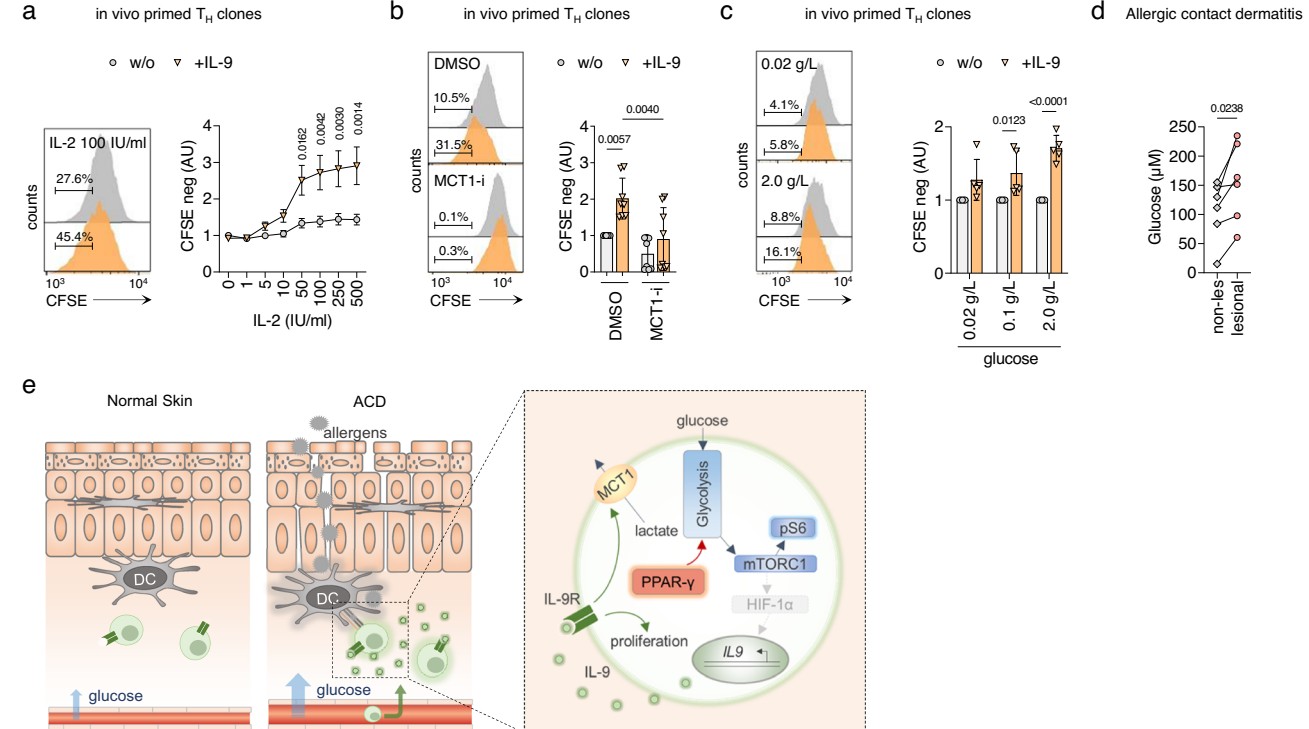

**Fig. 6 | IL-9 promotes T-cell proliferation in the high-glucose environment of allergic contact dermatitis. a** Proliferation of in vivo primed IL-9R⁺ TH clones in presence of IL-9 and different concentrations of IL-2, measured with CFSE dilution by flow cytometry after 3 days. **b** Proliferation of in vivo primed IL-9R⁺ TH clones in presence of IL-9 and MCT1 inhibitor (MCT1-i) BAY-8002 measured as in **a**. **c** In vivo primed IL-9R⁺ TH clones were cultured in media containing glucose of different levels for 7 days. Proliferation was measured in the presence of IL-9 with CFSE dilution by flow cytometry as in **a**. **d** Glucose concentrations measured with the Glucose-Glo™ Assay (Promega) of interstitial fluids of lesional skin of positive patch test reactions to different allergens (Supplementary Table S3) 48 h post allergen application and adjacent non-lesional skin biopsies. **e** Schematic presentation of the main conclusions. The mTORC1-HIF-1α-IL-9 axis has previously been established by others[24–26]. The data are representative of one experiment with four **a** or five **c** clones from one donor or independent experiments with eight clones from one donor **b** or independent experiments with six donors **d**. Statistics: **a, c** Two-way ANOVA, followed by a Šidák's test for multiple comparisons. **b** One-way ANOVA, followed by a Tukey's test for multiple comparisons. **d** Two-tailed paired *t* test. The data are presented as mean ± SD. Only *p* values <0.05 are shown.

Next, we investigated the expression and regulation of *SLC16A1* in different TH cell populations. In vitro TH9 differentiation induced higher levels of *SLC16A1* than TH1, TH2, or iTREG differentiation (Fig. 5i). Time course transcriptomics of TCR-stimulated TH clones[9] revealed a significantly higher expression of *SLC16A1* in TH9 clones than in TH1 and TH2 clones (Fig. 5j), as well as a close correlation between the expression of *IL9* and *SLC16A1* (Fig. 5k). Finally, a Seahorse experiment confirmed that IL-9R⁺ TH clones stimulated by IL-9 exhibited strongly elevated ECAR, in line with IL-9-dependent induction of active aerobic glycolysis and efficient cellular export of lactate (Fig. 5l). Accordingly, IL-9 increased extracellular lactate levels in cultured IL-9R⁺ TH clones, and these levels were suppressed by the addition of BAY-8002, a potent MCT1 antagonist (MCT1-i) or by siRNA-mediated *SLC16A1* knockdown (Fig. 5m). To link IL-9, glycolysis and MCT1 expression, we next investigated IL-9 levels in TH9 cells in presence of MCT1 inhibitor. We show that MCT1 inhibition reduces IL-9, but not IL-13 levels (Supplementary Fig. 5a). On the contrary, low-glucose environments, which in turn lead to reduced IL-9 levels, inhibited the induction of *SLC16A1* (Supplementary Fig. 5b).

Taken together, these data indicate that paracrine IL-9 promotes aerobic glycolysis in IL-9R⁺ TH cells by inducing the lactate transporter MCT1, which controls glycolytic flux.

### IL-9 promotes T cell proliferation in high-glucose environments

Considering the crucial role of aerobic glycolysis in supporting T cell proliferation[30], we next investigated the functional effects of IL-9-induced MCT1 expression on T cell proliferation. IL-9 induced strong proliferative responses in IL-9R⁺ TH clones (Fig. 6a), and this proliferative boost was reversed by adding BAY-8002, the MCT1 inhibitor (Fig. 6b). Moreover, IL-9-induced proliferation was dependent on available glucose levels (Fig. 6c), which further supports the notion that IL-9 promotes glycolytic flux in IL-9R⁺ TH cells to facilitate proliferation.

Finally, we investigated whether tissue glucose levels are dynamically regulated in acute allergic skin inflammation, in which TH9 cells have been shown to highly express IL-9[9,31]. To this end, we determined glucose levels in the interstitial fluid of tissue homogenates from non-lesional and lesional ACD skin 48 h post allergen application, respectively. The interstitial fluid of the lesional skin contained higher glucose levels than the matched non-lesional skin samples (Fig. 6d), which suggests that IL-9-related human allergic skin inflammation is associated with changes in glucose availability in vivo.

Collectively, these observations suggest that paracrine IL-9 facilitates the proliferation of T cells by stimulating aerobic glycolysis through the induction of the lactate transporter MCT1, possibly contributing to the proliferation of IL-9R⁺ TH cells in the high-glucose environment of ACD and acute allergic tissue inflammation. In addition, we found that PPAR-γ is a positive regulator of aerobic glycolysis in activated human TH9 cells, which in turn, regulates the expression of IL-9 via mTORC1. Together, this suggests that PPAR-γ and IL-9 facilitate immunometabolic sensing of the tissue microenvironment (Fig. 6e).

## Discussion

Here, we used in vivo and in vitro primed TH9 cells, which represent a subpopulation of PPAR-γ⁺ TH2 cells and share key characteristics with disease-associated pTH2 cells, to study the role of PPAR-γ in human TH

cells. We provide evidence that PPAR-γ is a positive regulator of aerobic glycolysis in activated human $T_H9$ cells. High glycolytic activity, in turn, was found to differentially regulate the effector function, particularly the expression of IL-9. Paracrine IL-9 signaling enhanced the expression of the lactate transporter MCT1, which provided a proliferative advantage to IL-9R$^+$ $T_H$ cells in high-glucose environments. Our data point to a previously unknown role of PPAR-γ and IL-9 in facilitating immunometabolic sensing of the tissue microenvironment. In the future, the PPAR-γ-mTORC1-IL-9 axis can be evaluated as a therapeutic target for type 2-driven skin inflammation and $pT_H2$-mediated disease.

*PPARG* is highly and specifically expressed in $pT_H2$ cells isolated from a variety of human type 2-driven diseases[4–10]. In mice, PPAR-γ is functionally important for the pathogenicity of $T_H2$ cells in allergic tissue inflammation, where it regulates the expression of IL-33R and increases the sensitivity of $T_H2$ cells to the tissue alarmin IL-33[11,12]. While IL-33R (*IL1RL1*) is also associated with the $pT_H2$ phenotype in humans[4,6–8,22], we did not find that PPAR-γ regulates its expression in $T_H9$ cells. However, we found that PPAR-γ promotes the expression of glycolytic enzymes in activated $T_H9$ cells, thereby supporting the crucial metabolic switch to aerobic glycolysis that early T cell activation and proliferation depend on[32,33]. Our findings align with a recent report of genome-wide CRISPR screens in $T_H$ cells, where PPARG, together with GATA3, IRF4 and BATF, was found to form a core $T_H2$ regulatory network[34]. In that analysis, PPAR-γ was identified as particularly important for the activation of $T_H2$ cells but to a lesser extent for their differentiation. This may be related to the role of PPAR-γ in supporting aerobic glycolysis in $pT_H2$ cells, as observed in the present study. In fact, both rapid upregulation of aerobic glycolysis[24] and sustained cell proliferation after activation[35] have been previously described in murine $T_H9$ cells. Our data now suggest that this is, at least in part, a PPAR-γ-dependent process, providing a functional explanation for the association of PPAR-γ with the $pT_H2$ phenotype. Functionally, PPAR-γ might thus provide $pT_H2$ cells with an advantage over neighboring cells in competition for critical nutrients, such as glucose, in acute allergic tissue inflammation.

PPAR-γ has previously been shown to be downstream of mTORC1 and to promote fatty acid uptake in activated $T_H$ cells[20]. In our study, however, mTORC1 activation was dependent on PPAR-γ activity and PPAR-γ-antagonism had no effect on FA uptake, neither in in vitro primed nor in in vivo primed $T_H9$ cells. The details of these discrepancies require further study.

Immunometabolism is increasingly appreciated as an essential regulator of the T cell effector function[30]. However, the full spectrum of effector functions regulated by glycolytic activity in $T_H$ cells remains incompletely understood[36]. Previously, we found a differential effect of PPAR-γ antagonism on $T_H2$ cytokine profiles, with IL-9 expression dependent on PPAR-γ-signaling while IL-13 being expressed independently thereof[9]. In the present study, we identified glycolytic activity and sequential mTORC1 activation as a mechanistic link between PPAR-γ and IL-9 expression. Indeed, mTORC1 has been shown to promote IL-9 expression in murine $T_H$ cells by activating HIF-1α, which has a direct binding site on the *IL9* promoter[24]. Therefore, our study further supports the notion that distinct metabolic pathways can have a selective effect on the $T_H$ cell effector function and, in principle, demonstrates that these immunometabolic links could be leveraged for targeted manipulation of immune function[37].

Our finding of increased interstitial glucose levels in acute skin inflammation is of particular interest, as the skin is generally considered a low-glucose environment[21]. A recent study in mice has also found increased tissue glucose levels in type 2-mediated lung inflammation and has linked increased glycolytic activity to pathogenic functions of type 2 innate lymphoid cells (ILC2)[38]. Thus, $pT_H2$ cells might be particularly adapted to function in the high-glucose environment of acute allergic tissue inflammation due to the high glycolytic capacity provided by PPAR-γ.

The specific dependence of IL-9 production on active glycolysis in $T_H9$ cells raises the question of why a defined immunological signal depends on the availability of a particular nutrient. The putative answer comes from our finding that IL-9 induces MCT1 expression, enabling efficient lactate export and NAD$^+$ regeneration for further glycolytic activity via GAPDH. Indeed, inhibition of MCT1 during T cell activation selectively inhibits early T cell proliferation[39] and has been proposed as a novel approach for immunosuppressive therapy[40]. IL-9 is a pleiotropic proinflammatory cytokine in allergic inflammation, but its target cells and mechanism of action remain not fully defined[41–43]. Our data now suggest a previously unrecognized role of IL-9 in inducing lactate transport capacity in IL-9R$^+$ $T_H$ cells, whereby it increases their glycolytic capacity and proliferative potential. Interestingly, *SLC16A1* expression is also upregulated in lesional skin of atopic dermatitis and forms part of a dynamic immune signature that reflects progressive immune activation dominated by $T_H2$ cells[44].

Our study has several limitations and raises intriguing questions that remain to be addressed. For example, it remains unknown to what extent the glucose concentrations used in our in vitro assays are representative of the in vivo metabolic environment. Experimental modulation of interstitial levels of ubiquitous metabolites, such as glucose, remains challenging even in animal models[45]. The most critical metabolic pathways for T cell function have been identified in reductionist in vitro assays. Furthermore, how $pT_H2$ cells compete with neighboring tissue cells for nutrients, such as glucose, remains to be addressed. It is worth investigating whether PPAR-γ indeed confers a competitive metabolic advantage to $pT_H2$ cells in vivo and whether such differences in metabolic fitness translate into functional outcomes that can be targeted therapeutically. Finally, it remains to be investigated whether the PPAR-γ-mTORC1-IL-9 axis and its downstream target MCT1 are viable therapeutic targets for allergic skin inflammation. Collectively, our findings encourage further research into the molecular details of how PPAR-γ regulates the metabolism and function of $pT_H2$ cells to enable the development of novel therapeutic interventions.

## Methods
### Study and experimental design
The aim of this study was to investigate the mechanism by which PPAR-γ regulates the effector function of human $T_H9$ cells, which share key characteristics with $pT_H2$ cells. We used in vitro and in vivo primed PPAR-γ$^+$ $T_H9$ cells and we performed RNA-seq analysis of activated $T_H9$ clones upon treatment with the PPAR-γ antagonist GW9662. This analysis pointed to a role of PPAR-γ in the regulation of glycolytic activity in $T_H9$ cells. This hypothesis was tested in vitro by measuring real-time extracellular acidification rate using Seahorse Analyzer as well as glucose uptake, proliferation, and cytokines expression by flow cytometry, in low and high-glucose environments and/or in the presence or absence of the PPAR-γ inhibitor. We next investigated the ability of mTORC1 to sense glucose availability and mediate the PPAR-γ-dependent IL-9 expression by assessing S6 phosphorylation on cell-based experiments but also on skin samples of ACD, on which immunofluorescence staining was performed. Further, we tested whether IL-9 regulates glucose metabolism and downstream effector functions in IL-9R$^+$ $T_H$ cells by performing RNA-seq in the presence or absence of IL-9. The analysis revealed that IL-9-stimulated $T_H$ cells showed an induction of genes involved in aerobic glycolysis, which was confirmed at the protein level. The effect of IL-9 in increasing glycolytic capacity and proliferative potential of IL-9R$^+$ $T_H$ cells was analyzed by Seahorse experiments and CFSE dilution using flow cytometry, respectively. Finally, to evaluate how tissue glucose levels are dynamically regulated in acute allergic skin inflammation, interstitial fluid of tissue homogenates from non-lesional and lesional skin of ACD were analyzed with

a glucose detection luminescence-based assay. In our cell-based experiments, at least three biological replicates were analyzed in each single experiment and PBMCs from different donors were used, as indicated in the figure legends. All experiments performed on human tissue samples were conducted in accordance with the Declaration of Helsinki. Human blood was obtained from healthy donors from the Swiss Blood Donation Center in Bern and was used in compliance with the Federal Office of Public Health (authorization no. P_149). The skin was obtained from healthy patients who underwent cosmetic surgery procedures or patients with ACD, or positive patch test reactions to standard contact allergens. The study on human patient samples was approved by the Medical Ethics Committee of the Canton of Bern, Switzerland (no. 088/13; 2019-01068; 2019-00803). Written informed consent was obtained from all patients. Mechanistic studies on blood and human tissue cells were performed using in vitro assays without blinding or randomization.

### Isolation of human peripheral blood mononuclear cells (PBMCs)
Peripheral blood mononuclear cells (PBMCs) were isolated according to the manufacturer's Standard Operating Procedure (SOP): PBMC Isolation using SepMate™ (Stemcell Technology).

### Generation of in vivo primed $T_H1$, $T_H17$, $T_H2$, and $T_H9$ clones
CD4$^+$ T cells were isolated from PBMCs using the EasySep™ Human CD4 positive selection kit II (Stemcell Technologies) as per the manufacturer's instructions. Positively selected CD4$^+$ T cells were stained for the subsequent sorting of the $T_H$ cell subset. Memory $T_H$ cell subsets were sorted with a purity of >90% according to the expression of chemokine receptors from CD45RA$^-$CD25$^-$CD8$^-$CD3$^+$ cells: $T_H1$ (CXCR3$^+$CCR8$^-$CCR6$^-$CCR4$^-$), $T_H2$ (CXCR3$^-$CCR8$^-$CCR6$^-$CCR4$^+$), $T_H9$ (CXCR3$^-$CCR8$^+$CCR6$^-$CCR4$^+$), and $T_H17$ (CXCR3$^-$CCR8$^-$CCR6$^+$CCR4$^+$) using the MoFlow ASTRIOS with Summit v. 6.3.1 software (Beckman Coulter) (Supplementary Fig. 6a). Individual memory $T_H$ cells were directly sorted from CD4$^+$ T cells into 96-well plates according to the expression of chemokine receptors (see Isolation and purification of human T cell subsets from peripheral blood). Individual cells were grown by periodic activation with phytohemagglutinin (1 μg/ml; Sigma-Aldrich), and irradiated allogeneic feeder cells (5 × 10$^4$ per well) in a culture medium. Half of the nutrient medium for T cell culture was replaced with a fresh medium every second day, starting from day 2 after reactivation. $T_H$ cell clones were analyzed in the resting state (≥14 days after the last expansion) or at different time points after polyclonal activation (see T cell culture and activation).

### Generation of in vitro primed $T_H1$, $T_H2$, $T_H9$, and $iT_{REG}$ cells
Human naive T cells were isolated from PBMCs using the EasySep™ Human naive CD4$^+$ T Cell Isolation Kit (Stemcell Technologies) as per the manufacturer's instructions. Naive T cells were stimulated with αCD3/CD2/CD28 beads (T cell/bead = 2:1, Miltenyi) and primed into effector CD4$^+$ T cell subsets with IL-12 (5 ng/ml) (BioLegend) for $T_H1$ cells, IL-4 (50 ng/ml) (BioLegend) for $T_H2$ cells, IL-4 (50 ng/ml) and TGF-β (5 ng/ml) (R&D Systems) for $T_H9$ cells, and TGF-β (5 ng/ml) for $iT_{REG}$. From cell culture initiation to analysis, the culture medium was supplemented with the indicated cytokines every other day. Cells were harvested for RNA sequencing (RNA-seq), quantitative reverse transcription-polymerase chain reaction (RT-qPCR) analysis, or intracellular cytokine analysis by flow cytometry at different time points (see below).

### T-cell culture and activation
The culture medium consisted of RPMI 1640 with Hepes (Gibco) supplemented with 5% heat-inactivated human serum (Swiss Red Cross, Basel, Switzerland), 1% Glutamax (Gibco), penicillin (50 U/ml) and streptomycin (50 μg/ml) (BioConcept), and IL-2 (50/250 IU/ml) (Roche). The glucose medium consisted of glucose-free RPMI 1640

(Gibco) supplemented with 5% heat-inactivated dialyzed FBS (Gibco), 1% Glutamax (Gibco), penicillin (50 U/ml) and streptomycin (50 μg/ml) (BioConcept), IL-2 (50 IU/ml) (Roche) and glucose (Sigma-Aldrich). The fatty acid-free medium consisted of PBS (pH 7.4) supplemented with 0.5% fatty acid-free BSA (Sigma-Aldrich) and 1 mM EDTA. T cells were cultured in a 96-well plate with a density of 0.25 × 10$^5$ to 1 × 10$^5$ cells per well in a total volume of 200 μl of cell culture medium. T cells were polyclonally activated using ImmunoCult Human CD3/CD2/CD28 T Cell Activator (1:100) (Stemcell Technologies).

### Isolation of human T cells from skin biopsies
Lesional and non-lesional skin biopsies of positive patch test reactions to different allergens were cultured in a culture medium as described above, and respective treatments were added to the culture. T cells were harvested and analyzed by flow cytometry and RNA-seq.

### Analysis of cytokine expression and S6 phosphorylation by flow cytometry
All antibodies used for flow cytometry are listed in Supplementary Table S1. To analyze cytokine production and S6 phosphorylation, T cells were polyclonally activated using ImmunoCult Human CD3/CD2/CD28 T Cell Activator (1:100) (Stemcell Technologies). Before activation and at different time points thereafter, T cells were additionally stimulated with PMA (50 ng/ml) (Sigma-Aldrich), ionomycin (1 μM) (Sigma-Aldrich), and brefeldin A (10 μg/ml) (Sigma-Aldrich) for 4 h. After viability and surface staining, the cells were fixed and permeabilized using Cytofix/Cytoperm kit (BD Biosciences) as per the manufacturer's instructions. Fluorescence-labeled antibodies were used to detect intracellular proteins, as well as phosphorylation. All antibodies used for flow cytometry are listed in Supplementary Table S1.

### Proliferation assays
Proliferation assays were performed with carboxyfluorescein diacetate succinimidyl ester (CFSE) staining or Phase-Flow™ FITC BrdU Kit (BioLegend). For the CFSE, assay cells were labeled with CFSE (2 μM) in PBS and incubated at 37 °C for 8 min. Staining was quenched by adding two times the initial staining volume of the cell culture medium and incubating the cells at 37 °C for 5 min. After 3 to 4 days, the stained cells were acquired on CytoFLEX (Beckman Coulter). The Phase-Flow™ FITC BrdU Kit was performed according to the manufacturer's instructions and acquired on CytoFLEX. The data from both assays were analyzed using CytExpert v.2.4 software (Beckman Coulter) or FlowJo v.10 software (BD Life Sciences).

### Western blotting
For the analysis of S6 and pAMPK phosphorylation, cells were harvested after treatment, washed with PBS containing the Halt™ Protease and Phosphatase Inhibitor cocktail (Thermo Scientific), and lysed in 20 mM Tris-HCl pH 7.5, 0.5% NP40, 25 mM NaCl, and 2.5 mM EDTA containing the Halt™ Protease & Phosphatase Inhibitor cocktail (Thermo Scientific). Protein concentration was measured using the Pierce BCA protein assay kit (Thermo Scientific). Samples (5–10 μg of protein per lane) were loaded onto 10% SDS-PAGE gel. For the Western blot analysis of MCT1, 1 × 10$^6$ cells were lysed in 30 μl of a sample loading buffer (SLB) consisting of 62.5 mM Tris-HCl (pH 6.8), 2% 2-mercaptoethanol, 2% SDS, 0.02% bromophenol blue, 14.8% Glycerol, and 6 M Urea. For the western blot analysis of PPAR-γ, protein extracts were prepared as follows. Cells were resuspended in an ice-cold lysis buffer (20 mM HEPES, pH 7.8, 10 mM KCl, 1 mM EDTA, 0.65% Nonidet P-40, 1 mM DTT, 1 mM PMSF) and incubated for 15 min on ice. Nuclei were pelleted at 20,000 × g for 20 min at 4 °C and lysed in 30 μl SLB. After electrophoresis (150 V, 45 min), proteins were transferred to a 0.45 μm Nitrocellulose Blotting membrane (Amersham™ Protran™) by wet transfer (100 V, 75 min). Non-specific sites were blocked for 1 h

with 5% non-fat milk in a TBS-T buffer (25 mM Tris, pH 7.5, 150 mM NaCl, and 0.1% Tween 20). Primary antibodies were incubated overnight at 4 °C. Membranes were washed with the TBS-T buffer and incubated for 1 h at room temperature with the corresponding secondary antibodies. Using Western Bright Quantum (Advansta) or SuperSignal™ West Atto Ultimate Sensitivity Substrate (Thermo Scientific) the binding of specific antibodies was visualized thereafter using Fusion Pulse TS of Vilber Lourmat with the Evolution Capt Pulse 6 v. 17.02 software (Witec). All antibodies used for Western blotting are listed in Supplementary Table S1. Uncropped and unprocessed scans of all Western blots are included in the source data file.

### RNA sequencing (RNA-seq)

Total RNA was isolated from T cells using the RNeasy kit (Qiagen) as per the manufacturer's instructions. The samples were submitted to the Next Generation Sequencing (NGS) Platform (Institute of Genetics, University of Bern). RNA integrity was analyzed by Qubit™. For all samples, the RNA Integrity Number (RIN) values were ≥8. The total RNA was transformed into a library of template molecules using TruSeq® Stranded mRNA Sample Preparation Kits (Illumina®) and the EpMotion 5075 (Eppendorf) robotic pipette system. Single-end 100 bp and paired-end 50 bp sequencing were performed using HiSeq3000 (Illumina®). The RNA-seq reads were mapped to the reference human genome (GRCh38, build 81) using HISAT2 v. 2.0.4[46]. HTseq-count v. 0.6.1[47] was used to count the number of reads per gene, and DESeq2 v.1.4.5[48] was used to test for differential expression between groups of samples. The RNA-seq data are deposited on BioStudios (accession no. E-MTAB-12204, E-MTAB-12199, E-MTAB-12237, E-MTAB-12197).

### Seahorse

The OCR and ECAR were measured 24 h after treatment using the Seahorse XFe96 Analyzer with the Wave v. 2.6.3 software (Seahorse Biosciences, Agilent Technologies) as per the manufacturer's instructions. On the day of the assay, the culture medium was replaced with the Seahorse XF base medium (catalog no. 102353-100), supplemented with reagents necessary to meet the cell culture conditions. Cells were seeded into Seahorse XFe96-well plates (Seahorse Biosciences, Agilent Technologies) with a density of $1.5 \times 10^5$ cells per well and a total volume of 50 µl of cell culture medium to obtain eight replicates per condition. Then, cells were equilibrated for 1 h in a non-$CO_2$ incubator at 37 °C. After measuring the baseline, successive injections of oligomycin (1 µM), carbonyl cyanide-p-trifluoromethoxy-phenylhydrazone (FCCP) (1 µM), rotenone (1 µM), antimycin A (1 µM), and 2-deoxy-d-glucose (2-DG) (50 mM) were delivered to measure the mitochondrial OCR and ECAR. The data were normalized to the DNA content which was determined after each Seahorse assay using CyQUANT™ Cell Proliferation Assay (Thermo Fisher Scientific) as per the manufacturer's instructions.

### Lactate measurements

$T_H9$ clones were pre-incubated with DMSO or BAY-8002 (75 µM) for 1 h before adding IL-9 (5 ng/ml) or culture medium only. After 48 h, the lactate levels were measured using the Lactate-Glo™ Assay (Promega) as per the manufacturer's instructions. Luminescence was measured using Tecan Reader Spark 10 M with the Tecan Spark Control v. 1.2 software (Tecan).

### Glucose measurements of interstitial fluid

Lesional skin biopsies from positive patch test reactions to different allergens (Supplementary Table S3) and adjacent non-lesional skin biopsies were taken 48 h after allergen application. No steroids or other products were applied to the skin before or during the patch test. Biopsies were weighed and resuspended in PBS. The volume of PBS corresponded to 8 times the weight of the biopsy. Skin biopsies were centrifuged at $300 \times g$ for 8 min to isolate interstitial fluid of biopsy samples. Glucose concentration in the interstitial fluid was measured using the Glucose-Glo™ Assay (Promega) as per the manufacturer's instructions. Luminescence was measured using Tecan Reader Spark 10 M with the Tecan Spark Control v. 1.2 software (Tecan).

### Glutamine and glucose measurements

Extracellular glutamine of $T_H9$ cells primed in vitro for 7 days in presence of GW9662 was measured in the supernatant using the Glutamine Assay Kit (Abcam) according to the manufacturer's instructions. At the same time, glucose levels were determined using the AccuCheck®.

### Glucose Uptake

2-NBDG was used as a tool to study cellular glucose uptake. Cells were washed in PBS and incubated with 1 ng/ml 2-NBDG (Cayman) in a glucose-free medium for 15 min at 37 °C in 5% $CO_2$. After washing, cells were either analyzed by flow cytometry or sorted into three different populations (low, medium, and high) according to their 2-NBDG uptake rate. RNA was isolated for the RT-qPCR analysis.

### Fatty acid uptake

BODIPY™ FL $C_{16}$ was used as a tool to study cellular fatty acid uptake. Cells were washed in PBS and incubated with 20 nM BODIPY™ FL $C_{16}$ (Thermo Fisher Scientific) in fatty acid-free medium for 15 min at 37 °C in 5% $CO_2$. After washing, cells were analyzed by flow cytometry.

### Immunofluorescence

Skin biopsies of ACD patients and normal skin biopsies were embedded in paraffin, cut into 4 µm thick sections, and heated for 20 min at 63 °C. The samples were stained using a BOND Autostainer and included dewaxing, pre-treatment with a buffer pH 9 for 20 min at 95 °C, and sequential double staining. Skin biopsies from positive patch test reactions and normal skin biopsies were placed in an optimal cutting temperature (OCT) compound, snap-frozen, and stored at −80 °C. Samples were cut into 6 µm cryo-sections, fixed with acetone at 4 °C for 10 min, and blocked with normal goat serum (1:50) (Dako) at room temperature for 15 min. Primary antibodies were added at room temperature for 60 min, followed by washing with TBS-Saponin 0.1%. Secondary antibodies were also added at room temperature for 60 min, followed by washing with TBS-Saponin 0.1%. Slides were mounted using Fluoromount-G™ Mounting Medium with DAPI (Southern Biotech). Immunofluorescence images were acquired on an Eclipse Microscope (Nikon) using the NIS Elements Imaging v. 4.2 software (Nikon). All antibodies used for Immunofluorescence are listed in Supplementary Table S1.

### Quantitative RT-qPCR

Total RNA was isolated from cultured in vitro or in vivo primed T cells or ex vivo sorted T cells using the RNeasy kit (Qiagen) as per the manufacturer's instructions. RNA from snap-frozen skin biopsies was isolated using the RNeasy Lipid Tissue Kit (Qiagen) as per the manufacturer's instructions. The total mRNA quality was measured using the ND-1000 Spectrophotometer (Thermo Fisher Scientific) or 2100 Bioanalyzer (Agilent). Complementary DNA was generated using Omniscript reverse transcriptase (Qiagen). Real-time PCR was performed using TaqMan probe-based assays and measured using the 7300 Real-Time PCR System (Applied Biosystems) and the Sequence Detection v. 1.4 software. The expression of each ligand transcript was determined relative to the reference gene transcript (HPRT-1) and normalized to the expression of the target gene using the $2^{-\Delta\Delta Ct}$ method. The data are represented as arbitrary relative units. All used primers were acquired from Thermo Fisher Scientific and are listed in Supplementary Table S2.

## Gene silencing by siRNA

For gene silencing of PPAR-γ by siRNA, in vitro primed $T_H9$ cells at day 6 were electroporated (4D-Nucleofector, Lonza: Buffer P3, pulse E0-115) for transfection with Silencer™ Select Negative Control No. 1 siRNA (Thermo Fisher Scientific) or three Silencer™ Select *PPARG* siRNAs (Thermo Fisher Scientific #s10886, #s10887, #s10888). After 24 h, the transfected cells were analyzed by RT-qPCR, and glucose uptake was measured. For gene silencing of *SLC16A1* by siRNA, in vivo primed $T_H9$ clones were electroporated (4D-Nucleofector, Lonza: Buffer P3, pulse E0-115) for transfection with (i) ™ Select Negative Control No. 1 siRNA (Thermo Fisher Scientific), (ii) two Silencer™ Select *SLC16A1* siRNAs (Thermo Fisher Scientific #s579, #s580) or (iii) MCT1 siRNA (Santa Cruz #sc-37235). After 12 h, transfected cells were analyzed by RT-qPCR and incubated with IL-9 (5 ng/ml) for 36 h. Extracellular lactate was measured as described above. For gene silencing of *RPTOR* by siRNA, in vitro primed $T_H9$ cells were electroporated (4D-Nucleofector, Lonza: Buffer P3, pulse E0-115) for transfection with Silencer™ Select Negative Control No. 1 siRNA (Thermo Fisher Scientific) or RPTOR siRNA (Santa Cruz #sc-44069). After 12 h, transfected cells were analyzed by RT-qPCR and stimulated with αCD3/CD2/CD28 for 24 h. Cytokines and pS6 were measured as described above. All siRNA used are listed in Supplementary Table S4.

## RNA-seq of $T_H$ cell subsets primed in vitro

Naive T cells were isolated from PBMCs by EasySep negative selection kit (Stemcell Technologies) according to the manufacturer's instructions. Naive $T_H$ cells were differentiated under $T_H1$ (IL-12), $T_H2$ (IL-4), $T_H9$ (IL-4+TGF-β), or $iT_{REG}$ (TGF-β) priming conditions for 7 days (see Materials and Methods, In vitro T cell differentiation). Total RNA was isolated from clones with the RNeasy Micro Kit (Qiagen) according to manufacturer's instruction and RNA-seq was performed.

## RNA-seq of non-lesional and lesional skin biopsies post allergen application

Six donor-matched skin biopsies were taken from non-lesional (NL) and lesional skin of positive patch test reactions to nickel at 24 h, 48 h, and 120 h post allergen application, respectively. Skin biopsies from positive patch test reactions were placed in an optimal cutting temperature (OCT) compound, snap-frozen, and stored at −80 °C. Total RNA was isolated from biopsies with the RNeasy Lipid Tissue Mini Kit (Qiagen) according to manufacturer's instruction and RNA-seq was performed.

## RNA-seq of $T_H9$ clones in the presence of the PPAR-γ inhibitor GW9662

Human CD4$^+$ T cells were isolated from PBMCs by EasySep™ Human naive CD4$^+$ T Cell Isolation Kit (Stemcell Technologies) according to the manufacturer's instructions. Positively selected CD4$^+$ T cells were stained for subsequent $T_H$ cell subset sorting (see Materials and Methods, Isolation and purification of human T cell subsets from peripheral blood). Total RNA was isolated from $T_H9$ clones after incubation with DMSO or GW9662 for 48 h followed by 12 h activation with anti-CD3/CD2/CD28 with RNeasy Micro Kit (Qiagen) according to manufacturer's instruction.

## RNA-seq of IL-9R$^+$ $T_H$ cells in presence of recombinant human IL-9

Human CD4$^+$ T cells were isolated from PBMCs by EasySep positive selection kit (Stemcell Technologies) according to the manufacturer's instructions. Positively selected CD4$^+$ T cells were stained for subsequent $T_H$ cell subset sorting (see Materials and Methods, Isolation and purification of human T cell subsets from peripheral blood). After T cell single-cell cloning, cells were screened for IL-9R expression by flow cytometry. Skin biopsies from positive patch test reactions were cultured in culture medium and CD4$^+$ T cells were isolated and

screened for IL-9R expression by flow cytometry. Five IL-9R$^+$ $T_H$ clones isolated from blood and three IL-9R$^+$ $T_H$ cells isolated from skin biopsies, respectively, were incubated in presence or absence of recombinant human IL-9 (5 ng/ml) for 12 h. Total RNA was isolated with the RNeasy Micro Kit (Qiagen) according to manufacturer's instruction and RNA-seq was performed.

## Statistical analysis

Statistical analysis was performed using the GraphPad Prism 9 software. For between-group comparisons, a one-way or two-way analysis of variance (ANOVA) was used, followed by pairwise comparisons for each group and either a Dunnett's, Tukey's, or Šidák's test to correct for multiple testing. The matched samples were analyzed using two-tailed paired $t$ tests or repeated-measures ANOVA whereas unmatched samples were analyzed with two-tailed unpaired $t$ test. The $n$ values and the respective statistical methods for individual experiments are indicated in the figure legends. For all statistical analyses, a 95% confidence interval and $p$ value <0.05 were considered significant.

## Reporting summary

Further information on research design is available in the Nature Portfolio Reporting Summary linked to this article.

## Data availability

The RNAseq data of this study have been deposited in ArrayExpress with the accession codes E-MTAB-12204, E-MTAB-12199, E-MTAB-12237 and E-MTAB-12197. Publicly available data with accession code GSE93219, GSE130148, GSE175930 and E-MTAB-5739 were re-analyzed. The RNAseq reads were mapped to the reference human genome GRCh38, build 81 [https://www.ncbi.nlm.nih.gov/assembly/GCF_000001405.26]. The authors declare that all other data supporting the finding of this study are available within the article and its supplementary information files. Source data are provided with this paper.

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

## Acknowledgements

We thank the next-generation sequencing platform and the flow cytometry and cell sorting core facility of the Department of BioMedical Research, University of Bern, for performing the high-throughput sequencing experiments and the FACS sorts, respectively. We are grateful to Ben Roediger and Curdin Conrad for their helpful comments. This study was supported by the Peter Hans Hofschneider Professorship for Molecular Medicine, Swiss National Science Foundation (grant no. 320030_192479), Bern Center for Precision Medicine (Pilot Project grant), and Ruth & Arthur Scherbarth Foundation (project grant) (all to C.Sc.).

## Author contributions

N.L.B., O.S., F.L., L.v.M., C.B., S.S., A.V., N.B., and C.Sc. designed and performed the experiments; O.F., S.F., A.F., JM. N., M.L., and C.Sc. provided essential reagents and funding; S.R.H., M.P.G, C.L., M.B., D.S., and C.Sc. provided patient samples; N.L.B., F.L., O.S., L.v.M., I.K., and C.Sc. analyzed the data; N.L.B., F.L., and C.Sc. wrote, edited, and revised the article.

## Competing interests

The authors declare no competing interests. N.L.B., F.L., C.B., S.S., and C.Sc. are members of the SKINTEGRITY.CH collaborative research project.
