## [Peer Review File · Nature Communications]

PPAR- γ regulates the effector function of human T helper 9 cells by promoting glycolysisREVIEWER COMMENTS

Reviewer #1 (expertise in immunometabolism, CD4+ T cell metabolism):

In the paper by Bertschi and colleagues, the authors analyzed the role of peroxisome proliferator-activated receptor gamma (PPAR- γ) transcription factor in the regulation of the human T helper (Th)-9 cell effector functions. Specifically, they found that PPAR- γ controls activation-induced glycolysis, which, in turn, specifically promotes the expression of IL-9 in an mTORC1-dependent manner. The authors corroborated their findings on skin samples from subjects with allergic contact dermatitis also showing that IL-9 induced the lactate transporter MCT1 expression, responsible of the increased glycolysis and proliferative capacity of Th9 cells.

The paper is interesting and the topic is original as PPAR- γ , IL-9, and their downstream targets might represent novel therapeutic targets to modulate allergic contact dermatitis (ACD) as well as other Th2-driven diseases.

Although the manuscript is quite clear and well structured, there are some pitfalls that partially limit the enthusiasm for its publication, as listed below:

Major issues:

- 1) The first aspect concerns the methodological choice of using drugs to block different molecules (eg. Rapamycin, or GW9662, MCT inhibitor), with a pharmacological intervention. This approach can not exclude the possible off-target effects, also linked to the toxic effects of the used drugs. Therefore, it would be advisable to confirm the data also by inhibiting the expression of different molecules with specific silencing (ie. by shRNA) (genetic approach), in order to exclude side effects of the drugs, and more importantly, to be sure that the observed effects are specific and selective.
- 2) The authors evaluated the impact of PPAR- γ in the modulation of Th9 cells only by inhibiting this transcription factor (by pharmacological intervention). The authors should prove that, on the contrary, PPAR- γ overexpression is able to reverse the observed phenomena. These data would provide a more direct evidence of PPAR- γ 's involvement in the control of Th9 cell function.
- 3) The authors focused their attention on the role of PPAR- γ in the induction and control of glycolysis. However, it would be advisable also to show the effects of PPAR- γ modulation on the other pathways used for cellular energy support, such as mitochondrial respiration, glutaminolysis and fatty acid the oxidation. Is glycolysis the preferential pathway used by Th9 cells to produce energy? Do Th9 cells also use other intracellular metabolic pathways? All these aspects should be better investigated and discussed.
- 4) An aspect that deserves a deeper attention is the study of the molecular mechanism through which the PPAR- γ -mTOR-IL9 axis can regulate the expression of the lactate transporter MCT1. How does the regulation of its expression take place? is it a transcriptional regulation at the promoter level? through which mechanism does this process occur? Which factor is involved in this process? further details should be provided.
- 5) To link IL-9 with glycolysis and MCT1 expression, the authors should show the levels of IL-9 in the presence of MCT1 inhibitor and also, on the contrary, evaluate MCT1 expression when glycolysis is blocked. These experiments would provide a direct and univocal evidence of the role of PPAR- γ -mTOR-IL9 axis on Th9 cell function.

Minor issues:

- 1) In addition to 2-DG, the authors should use and confirm their data also with other glycolysis inhibitors, acting on different molecules (ie. Lonidamine).
- 2) Immunofluorescence images of dermatological lesions from allergic contact dermatitis (Figure 3F and H) should be compared to non-pathological conditions to better appreciate the difference, if any. Furthermore, those images should be quantified, in order to obtain a more objective estimation of the observed differences.
- 3) Cytofluorimetric histogram showing the CFSE staining (Figure 2I) is very confusing. From the histogram it is very hard to observe a proper dilution of CFSE dye. How do the authors comment and interpret this data?

Reviewer #2 (expertise in TH cell subsets, TH9 cells):

Since their characterization in 2008, Th9 cells have been shown to promote tissue inflammation in both mice and humans. Th9 cells shape disease course in multiple settings, including inflammatory and allergic diseases. While the transcription factor PU.1 has been identified as a major driver for IL-9 production for mouse and human CD4 T cells, most of the in vivo evidence showing the contribution of Th9 cells to disease stems from mouse studies. In this regard, the work presented in this manuscript outlines a very interesting signaling pathway contributing to CD4 T cell-derived IL-9 secretion in human skin inflammation. This study has many strengths:

- 1) It builds on previous mouse studies (notably PMID: 27317260) to extend to a human context the role of glycolysis and mTORC1 signaling in the secretion of IL-9 from Th9 cells.
- 2) It demonstrates the functional role of PPAR-g in driving glycolysis in human Th9 cells.
- 3) It mechanistically uncouples the secretion of IL-9 and IL-13 from effector Th9 cells.
- 4) It ascribes a novel role for IL-9 in enhancing the glycolytic capacity of IL-9R+ T cells

Overall, these findings will further underscore the relevance of the Th9 cell subset in tissue inflammation. The data presented are for the most part convincing and logically build on the previous work from the authors published in *Science Immunology*.

While the data focusing on Th9 cell biology are straightforward, some sections of the manuscript need to be clarified to better integrate all relevant findings in the field. My concerns are as follows:

- 1) some sections of the manuscript are unclear and even conflicting with the message conveyed in the abstract. For instance, to begin their Materials and Methods section, the authors state that their goal was to study human pTH2 cells. Th9 cells are then described as a model to understand "pTh2" cells. Does this mean that the authors regard "pTh2" cells as a distinct CD4 T cell subset (as compared to conventional Th2 cells)? Would this mean that Th9 cells are not a distinct CD4 T cell subset? If the authors believe this, it is then somewhat odd that they rely in the present work on Th9 cells. Defining conditions to generate pTh2 cells in vitro would have been more appropriate. More importantly, while I understand that the authors are focusing on PPARg, it would be fair to discuss, and experimentally document in Figure 1, the RNA and protein levels of PU.1 considering previous literature (for instance: PMID: 20431622). This would also be important because in their *Science Immunology* paper, the authors actually reported higher PU.1 expression levels in Th9 cells as compared to Th2 cells at early time points during differentiation.
- 2) In line with the previous comment, while I understand the authors' willingness to discuss in detail in the introduction the concept they have written about (reference 2), I think additional balance would be beneficial. Except for my comment on PU.1, I concur that Figure 1 strongly suggests a very close proximity between pTh2 cells and Th9 cells. Surprisingly, despite the presented data, the authors do not discuss this here while they somehow addressed this in reference 2. Should pTH2 cells be renamed Th9 cells (or the opposite)? Or do the authors think that Th9 cells non-responsiveness to IL-33 is enough to discriminate pTh2 and Th9 cells, especially considering that some laboratories reported that Th9 cells respond to IL-33 (see for instance PMID: 29038366)? Presenting all perspectives would be very important to clarify the field and present a balanced view.

In summary, this is a very interesting study whose impact could be further enhanced by clearly underscoring the relevance of Th9 cells instead of representing them as a useful tool to study a larger group of inflammatory cells. This is actually what the authors have done in their abstract and graphical abstract, leading to very clear and convincing claims.

Reviewer #3 (expertise in allergic inflammation, pathogenic TH cells):

Bertschi et al. showed the pathological significance of PPAR γ in helper T cell 9 (TH9) cells by performing the functional analysis of PPAR γ using in vitro primed TH9 cells and TH9 cells derived from patients with allergic contact dermatitis.

First, the authors showed that TH9 cells expressed the core feature genes of pathogenic TH2 cells, including enhanced expression of PPAR- γ , IL-5, IL-9, and IL-9R. They showed that glucose uptake in TH9 cells was PPAR- γ dependent. Furthermore, they found that the production of IL-9 and IL-13

by TH9 cells was associated with the glycolytic activity. They also found that glucose and PPAR- γ -dependent production of IL-9 in TH9 cells was regulated via mTORC1. Finally, they showed that IL-9 promotes aerobic glycolysis in IL-9R+ TH cells by inducing the lactate transporter MCT1. This study demonstrates the importance of the PPAR- γ -mTORC1-IL-9 axis in Th9 cells and is expected to aid in our understanding the type-2-driven skin inflammation. However, additional experiments are needed to support the authors' conclusions. Specific comments are described below.

Major comments:

1. The authors addressed the underlying molecular mechanisms by performing experiments of loss-of-function using inhibitors such as GW9662 and MCT1-i. The authors should perform the experiments of gain-of-function, such as the overexpression of PPAR- γ , mTORC1, or MCT1 to confirm the molecular mechanisms that they found.

2. In the introduction part, the authors cited previous their-own review paper to introduce pathogenic Th2 cells. But there are other comprehensive review article regarding pathogenic Th2 cells such as Nakayama et al *Annu Rev Immunol* 2017. The concept of pathogenic Th2 cells was originally proposed by this group, and the paper should be included. Furthermore, in the discussion section, they mentioned the roles of IL-33-IL-33R pathway in both human and mouse pathogenic Th2 cells without the citation of appropriate original article, Endo et al *Immunity* 2015. They must include appropriate previous work.

3. The authors mentioned that mTORC1 activation induces IL-9 expression via HIF-1 α in their Graphical abstract based on previous studies. But they did not present any data related to HIF-1 α in the present study.

4. In Fig. 1G, the expression of IL9 seems to be significantly lower than that of IL5 and IL13. They should investigate the expression of IL9 expression by performing other experiments such as real-time quantitative PCR. The reviewer also wondered whether the expression of PPAR- γ was upregulated in this experimental setting.

5. Since the authors have shown that TH9 cells specifically express high level of IL5 in Fig. 1C, they should address the production of IL-5 in Fig. 3A-C.

6. In Fig. 3C, the expression of PPARG should also be analyzed to investigate the activation of the glycolysis by PPAR- γ .

7. Regarding Fig. 4F and S4D, the authors should explain why they stained for CD3 or CD4 instead of the TH9 cell markers; CCR8 and CCR4 used in Fig. 2H seem to be more useful for markers to detect TH9 cells.

8. In Fig. 5C, immune-histological sample revealed that many CD3-negative IL-9R-expressing mononuclear cells were accumulated around CD3-positive T cells in the local tissues of allergic contact dermatitis. This result raised the possibility that IL-9 by TH9 cells may affect IL-9 receptor expressing CD3-negative cells, which consequently affect T cell function and glycolytic activity. To address this point, further experiments are required using in vivo primed TH9 clones, to measure the effects of IL-9 signal blockade using neutralizing antibodies against IL-9 or IL-9R blockade on glycolysis and MCT1 expression.

9. In Fig. 6D, how many hours after the start of the patch test were the glucose concentrations in the ACD model? The time course was clear in Fig. 1G, but not in this one. Also, were there any steroids or other products applied to the skin before or after the patch test?

Minor comments:

1. In Fig. 1E, the authors performed experiments using in vivo primed Th clones for the first time in this paper. To make it easier for readers to understand, they should add a brief explanation of the experimental system and describe it clearly in the Methods section.

2. In Fig. 2C and 2E, the authors showed the maximal glycolytic capacity as the graph. The graph showing the glycolysis rate before addition of Oligomycin should also be presented.

3. Fig. S3 seems to be missing.

4. Regarding Fig. 4 B and C, in lines 175-176, the authors stated "Indeed, IL-9+ T cells were strongly enriched in the pS6+ cell population, whereas the proportion of IL-13+ T cells was similar in the pS6- and pS6+ populations," but in order to make this claim, the proportion of IL-9+ and IL-13+ cells should be shown.

RESPONSE TO REVIEWERS' COMMENTS

Please find below the point-by-point responses to the reviewers' questions.

Changes in the manuscript are marked in blue.

1. Reviewer

In the paper by Bertschi and colleagues, the authors analyzed the role of peroxisome proliferator-activated receptor gamma (PPAR- γ) transcription factor in the regulation of the human T helper (Th)-9 cell effector functions. Specifically, they found that PPAR- γ controls activation-induced glycolysis, which, in turn, specifically promotes the expression of IL-9 in an mTORC1-dependent manner. The authors corroborated their findings on skin samples from subjects with allergic contact dermatitis also showing that IL-9 induced the lactate transporter MCT1 expression, responsible of the increased glycolysis and proliferative capacity of Th9 cells.

The paper is interesting and the topic is original as PPAR- γ , IL-9, and their downstream targets might represent novel therapeutic targets to modulate allergic contact dermatitis (ACD) as well as other Th2-driven diseases. Although the manuscript is quite clear and well structured, there are some pitfalls that partially limit the enthusiasm for its publication, as listed below.

Major issues:

1.1 The first aspect concerns the methodological choice of using drugs to block different molecules (eg. Rapamycin, or GW9662, MCT inhibitor), with a pharmacological intervention. This approach cannot exclude the possible off-target effects, also linked to the toxic effects of the used drugs. Therefore, it would be advisable to confirm the data also by inhibiting the expression of different molecules with specific silencing (ie. by shRNA) (genetic approach), in order to exclude side effects of the drugs, and more importantly, to be sure that the observed effects are specific and selective.

Response: Thank you for this valuable suggestion. We have now included data with siRNA targeting i) *PPARG*, ii) *RPTOR* or iii) *SLC16A1* that confirm our data obtained with the pharmacological inhibitors GW9662, rapamycin and BAY-8002, respectively.

i) We show that silencing of *PPARG* leads to decreased glucose uptake, whereas fatty acid (FA) uptake is not affected. These results confirm our findings obtained with the PPAR- γ inhibitor GW9662 (new Fig. 2g and new Supplementary Fig. 2h, adapted / new text, results (line 141 and 151-152)).

- ii) We performed siRNA knockdown of the mTORC1 protein *RPTOR*. Efficiency and specificity of *siRPTOR* knockdown was validated by detection of reduced phosphorylation of S6. We then found that IL-9 levels are significantly reduced in knockdown cells, whereas IL-13 levels are unaffected, thus corroborating our results obtained with rapamycin (new Supplementary Fig. 4d, adapted text, results (line 197)).
- iii) We show that successful silencing of *SLC16A1* by siRNA suppresses extracellular lactate levels in cultured IL-9R⁺ T_H clones in the presence of IL-9, but not in cells transfected with control siRNA. This is consistent with our data obtained with the MCT1 inhibitor BAY-8002 (new Fig. 5m, adapted text, results (line 240)).

Changes to the revised manuscript:

- New Fig. 2g: Glucose uptake of siRNA-induced *PPARG* knockdown in T_H9 cells
- New Supplementary Fig. 2h: FA uptake of siRNA-induced *PPARG* knockdown in T_H9 cells
- Adapted text, results (line 141): "Importantly, glucose uptake in T_H9 cells was reduced by PPAR- γ -inhibition or by siRNA-induced *PPARG* knockdown (Fig. 2g)."
- New text, results (line 151-152): "PPAR- γ antagonism had no effect on FA uptake (Supplementary Fig. 2h,i)."
- New Supplementary Fig. 4d: pS6 levels and cytokine expression of siRNA-induced *RPTOR* knockdown in T_H9 cells
- Adapted text, results (line 197): "Moreover, inhibition of mTORC1 by either siRNA against *RPTOR* or by rapamycin decreased the production of IL-9 but not IL-13 in activated T_H9 cells (Fig. 4d, e and Supplementary Fig. 4c, d)."
- New Fig. 5m: Extracellular lactate levels of siRNA-induced *SLC16A1* knockdown in T_H9 cells
- Adapted text, results (line 240): "Accordingly, IL-9 increased extracellular lactate levels in cultured IL-9R⁺ T_H clones, and these levels were suppressed by the addition of BAY-8002, a potent MCT1 antagonist (MCT1-i) or by siRNA-mediated *SLC16A1* knockdown (Fig. 5m)."

- 1.2 The authors evaluated the impact of PPAR- γ in the modulation of Th9 cells only by inhibiting this transcription factor (by pharmacological intervention). The authors should prove that, on the contrary, PPAR- γ overexpression is able to reverse the observed phenomena. These data would provide a more direct evidence of PPAR- γ 's involvement in the control of Th9 cell function.

Response: Thank you for this suggestion. Since overexpression PPAR- γ is not sufficient to increase PPAR- γ signaling, as retinoid X receptor (RXR) as well as (the mostly unknown) endogenous ligands are also

required, we decided to use the PPAR- γ agonist troglitazone (TGZ) in order to simulate PPAR- γ overexpression. However, unexpectedly, we did not detect higher IL-9 levels in presence of TGZ. Instead, we observed lower IL-9 levels, similar to the results obtained with the PPAR- γ antagonist GW9662. In parallel, we observed that S6 phosphorylation was reduced (new Supplementary Fig. 4e).

Previous studies have shown that PPAR- γ agonists such as TGZ activate AMPK¹. We therefore hypothesized that AMPK activation by TGZ leads to mTORC1 inhibition that in turn decreases IL-9 levels. Indeed, using Western blot analysis, we show that TGZ leads to phosphorylation of AMPK and inactivation of mTORC1 (new Supplementary Fig. 4f). Moreover, the AMPK activator A769662 also reduces IL-9, but not IL-13 levels (new Supplementary Fig. 4g).

Although we did not show that overexpression of PPAR- γ leads to a reversal of the phenotype that we observed with PPAR- γ inhibition, the additional TGZ / AMPK activation data supports our hypothesis that IL-9 expression is regulated via the mTORC1 axis. These novel insights are included into the manuscript (new text, results (line 199-206)).

Changes to the revised manuscript:

- New Supplementary Fig. 4e: Cytokine expression and pS6 levels in presence of TGZ in T_H9 cells
- New Supplementary Fig. 4f: Western blot analysis of pS6 and pAMPK in presence of TGZ in T_H9 cells
- New Supplementary Fig. 4g: Cytokine expression in presence of A769662 in T_H9 cells
- New text, results (line 199-206): "Since the PPAR- γ agonist Troglitazone (TGZ) unexpectedly reduced IL-9 expression in T_H9 cells, we next investigated the mechanism by which this occurs. Interestingly, pS6 levels were reduced in presence of TGZ (Supplementary Fig. 4e), suggesting that mTORC1 is inhibited. Previous studies have shown that PPAR- γ agonists, such as TGZ, activate AMP-activated protein kinase (AMPK)¹. We therefore hypothesized that TGZ-mediated AMPK activation negatively regulates mTORC1, which in turn suppresses IL-9 expression. Indeed, Western blot analysis revealed that TGZ leads to phosphorylation of AMPK and mTORC1 inhibition (Supplementary Fig. 4f). In addition, the AMPK activator A-769662 also reduced IL-9, but not IL-13 levels (Supplementary Fig. 4g). Together, this data strongly supports our hypothesis that IL-9 expression is mTORC1-dependent."

- 1.3 The authors focused their attention on the role of PPAR- γ in the induction and control of glycolysis. However, it would be advisable also to show the effects of PPAR- γ modulation on the other pathways used for cellular energy support, such as mitochondrial respiration, glutaminolysis and fatty acid the oxidation. Is glycolysis the preferential pathway used by Th9 cells to produce energy? Do Th9 cells also use other intracellular metabolic pathways? All these aspects should be better investigated and discussed.

Response: We thank the reviewer for this comment. As suggested, we performed additional experiments to study the effects of PPAR- γ inhibition on i) mitochondrial respiration, ii) fatty acid oxidation and iii) glutaminolysis.

- i) To address the effects of two different PPAR- γ antagonists on mitochondrial respiration, we now show the data for oxygen consumption rate (OCR) measurements as measured by Seahorse flux analysis (new Supplementary Fig. 2c, d, adapted text, results (line 135-136; 157)). In accordance with the current literature (reviewed in²) our data show that glycolysis is the dominant pathway engaged by activated T cells in high glucose environments, whereas mitochondrial respiration is only upregulated minimally. We therefore did not focus further on mitochondrial respiration because it is not the major metabolic pathway in the settings we studied (i.e. the conditions under which IL-9 is produced).
- ii) Since PPAR- γ has been implicated in mediating fatty acid uptake in activated T_H cells^{3,4}, we examined fatty acid (FA) uptake in different T helper subsets. However, in contrast to glucose uptake, we found that T_{H9} cells did not exhibit higher FA uptake compared with T_{H1}, and T_{H2} cells (new Supplementary Fig. 2g). Further, PPAR- γ antagonism had no effect on FA uptake of in vitro- or in vivo-primed T_{H9} cells (new Supplementary Fig. 2h, i). To investigate the contribution of FA metabolism to the regulation of cytokines in T_{H9} cells, we analyzed IL-9 expression after inhibition of FA oxidation. Culturing of T_{H9} clones in FA-free medium did not affect IL-9 or IL-13 expression (new Supplementary Fig. 3f), nor did inhibition of FA metabolism with etomoxir, an inhibitor of carnitine palmitoyltransferase-1 (new Supplementary Fig. 3g). This data is now included and discussed in our manuscript (new text, results (line 151-154; 158; 180-183); new text, discussion (line 291-294)).
- iii) To address the effect of PPAR- γ on glutaminolysis, we analyzed glutamine uptake in vitro primed T cells in presence of PPAR- γ blockade. We show that glutamine uptake is not meaningfully affected by PPAR- γ inhibition and hence conclude that glutaminolysis is not primarily regulated by PPAR- γ (new Supplementary Fig. 2j) in the settings studied here. In the results section of the manuscript, we added a corresponding statement (line 154-155; 158).

Changes to the revised manuscript:

- New Supplementary Fig. 2c, d: OCR measured by Seahorse flux analysis of T_{H9} cells cultured in media of different glucose levels in presence of GW9662 and T0070907, respectively.
- Adapted text, results (line 135-136): “We then performed measurements of oxygen consumption rate (OCR) and extracellular acidification rate (ECAR) in real-time before and after activation with α CD3/CD2/CD28 in either low or high-glucose environments.”

- Adapted text, results (line 157): “Moreover, in the setting studied here, ...”
- New Supplementary Fig. 2g: FA uptake of in vitro primed T_H cell subsets
- New Supplementary Fig. 2h, i: Effect on PPAR-γ antagonism on FA uptake in T_H9 cells by siRNA-mediated *PPARG* knockdown or by GW9662, respectively.
- New text, results (line 151-154): “Since PPAR-γ has been implicated in mediating fatty acid (FA) uptake in activated T_H cells^{3,4}, we examined FA metabolism in response to PPAR-γ antagonism. In vitro and in vivo primed T_H9 cells did not exhibit higher FA uptake compared with T_H1, and T_H2 cells (Supplementary Fig. 2g) and PPAR-γ antagonism had no effect on FA uptake (Supplementary Fig. 2h,i).”
- New text, results (line 158): “...whereas FA oxidation and glutaminolysis remain unaffected.”
- New Supplementary Fig. 3f: Cytokine expression of T_H9 cells cultured in medium without FA.
- New Supplementary Fig 3g: Cytokine expression of T_H9 cells in presence of etomoxir.
- New text, results (line 180-183): “To investigate the contribution of FA metabolism to the regulation of cytokines in T_H9 cells, we analyzed IL-9 expression in response to FA inhibition. Neither did culturing of T_H9 clones in FA-free medium affect IL-9 or IL-13 expression (Supplementary Fig. 3f), nor did inhibition of FA metabolism with etomoxir, an inhibitor of carnitine palmitoyltransferase-1 (Supplementary Fig. 3g).”
- New text, discussion (line 291-294): “PPAR-γ has previously been shown to be downstream of mTORC1 and to promote fatty acid uptake in activated T_H cells³. In our study, however, mTORC1 activation was dependent on PPAR-γ activity and PPAR-γ-antagonism had no effect of on FA uptake, neither in in vitro primed nor in in vivo primed T_H9 cells. The details of these discrepancies require further study.”
- New Supplementary Fig. 2j: Extracellular glutamine and glucose levels in T_H9 cells in presence of GW9662.
- New text, results (line 154-155): “In addition, glutamine uptake of T_H9 was not affected by PPAR-γ inhibition, suggesting that glutaminolysis is not primarily regulated by PPAR-γ in these cells (Supplementary Fig. 2j)”.

1.4 An aspect that deserves a deeper attention is the study of the molecular mechanism through which the PPAR-γ -mTOR-IL9 axis can regulate the expression of the lactate transporter MCT1. How does the regulation of its expression take place? is it a transcriptional regulation at the promoter level? through which mechanism does this process occur? Which factor is involved in this process? further details should be provided.

Response: Thank you for this comment. We have addressed this question in more detail and now show that the upregulation of the lactate transporter *SLC16A1* by IL-9 is JAK3 dependent (new Fig. 5h). We suggest that IL-9R signaling via JAK3 leads to the phosphorylation of STAT3 and STAT5, which then translocate to

the nucleus and regulate the transcription of *SLC16A1*. The result section in the manuscript is changed accordingly (line 231-232).

Changes to the revised manuscript:

- New Fig. 5h: *SLC16A1* expression in presence of IL-9 and the JAK3-inhibitor (JAK3-i) ritlecitinib
- New text, results (line 231-232): "In contrast, inhibition of JAK3, which is central to IL-9R signal transduction⁵, by ritlecitinib suppressed IL-9-induced upregulation of *SLC16A1* (Fig. 5h)."

1.5 To link IL-9 with glycolysis and MCT1 expression, the authors should show the levels of IL-9 in the presence of MCT1 inhibitor and also, on the contrary, evaluate MCT1 expression when glycolysis is blocked. These experiments would provide a direct and univocal evidence of the role of PPAR- γ -mTOR-IL9 axis on Th9 cell function.

Response: We performed the suggested experiments. We now include data showing that IL-9 expression is significantly reduced in the presence of the MCT1 inhibitor BAY-8002, whereas IL-13 expression is unaffected (new Supplementary Fig. 5a). In addition, we now show that *SLC16A1* expression is reduced in low glucose environments (new Supplementary Fig. 5b). These additional data are in line with the model proposed in our manuscript. The result section has been modified accordingly (line 240-243).

Changes to the revised manuscript:

- New Supplementary Fig. 5a: Cytokine expression of T_H9 cells in presence of the MCT1 inhibitor BAY-8002.
- New Supplementary Fig. 5b: *SLC16A1* expression in low and high glucose environment before and after activation with α CD3/CD2/CD28
- New text, results (line 240-243): "To link IL-9, glycolysis and MCT1 expression, we next investigated IL-9 levels in T_H9 cells in presence of MCT1 inhibitor. We show that MCT1 inhibition reduces IL-9, but not IL-13 levels (Supplementary Fig. 5a). On the contrary, low glucose environments, which in turn lead to reduced IL-9 levels, inhibited the induction of *SLC16A1* (Supplementary Fig. 5b)."

Minor issues:

1.6 In addition to 2-DG, the authors should use and confirm their data also with other glycolysis inhibitors, acting on different molecules (ie. Lonidamine).

Response: As suggested, we performed experiments with the glycolysis inhibitor lonidamine (LND). We show that LND reduces IL-9 and IL-5 levels but not IL-13 expression, confirming our data obtained in low

glucose and with 2-DG. Data are now included in the new Supplementary Fig. 3e and the text in the manuscript is adapted accordingly (line 174-175).

Changes to the revised manuscript:

- New Supplementary Fig. 3e: Cytokine expression of T_H9 cells in presence of LND.
- Adapted text, results (line 174-175): "In a next step, we hence inhibited glycolysis in T_H9 cells using the glucose analog 2-deoxy-d-glucose (2-DG) and the aerobic glycolysis inhibitor lonidamine (LND) to investigate the effect on cytokine expression. In T_H9 cells primed in vivo, 2-DG and LND inhibited the expression of IL-9 and IL-5 but not IL-13 (Fig. 3d and Supplementary Fig. 3d, e)."

- 1.7 Immunofluorescence images of dermatological lesions from allergic contact dermatitis (Figure 3F and H) should be compared to non-pathological conditions to better appreciate the difference, if any. Furthermore, those images should be quantified, in order to obtain a more objective estimation of the observed differences.

Response: We thank the reviewer for this suggestion. We now show additional immunofluorescence images of normal skin (NS) and quantified CD4⁺pS6⁺, PPAR-γ⁺ and PPAR-γ⁺pS6⁺ cells in ACD and NS, respectively. The new data is included in in Fig. 4 f, h and the new Supplementary Fig. 4i, k. The text in the manuscript is adapted accordingly (line 208-210; 214-215).

Changes to the revised manuscript:

- Adapted Fig. 4f and 4h: Quantification of CD4⁺pS6⁺ and PPAR-γ⁺pS6⁺ cells in ACD and NS, respectively
- New Supplementary Fig. 4i and k: Immunofluorescence images of NS stained for CD4 and pS6
- New Supplementary Fig. 4k: Immunofluorescence images of NS stained for PPAR-γ and pS6 as well as quantification of PPAR-γ⁺ cells in ACD and NS.
- Adapted text, results (line 208-210): "...we performed immunofluorescence staining of normal skin (NS) and ACD skin samples and isolated T cells from such lesions. Double immunofluorescence revealed that CD3⁺ and CD4⁺ T_H cells that express pS6 are significantly enriched in the infiltrate of ACD compared to NS (Fig. 4f and Supplementary Fig. 4h,i)."
- Adapted text, results (line 214-215): "Indeed, PPAR-γ⁺pS6⁺ double-positive lymphocytes were significantly increased in the dermis of ACD compared to NS (Fig. 4h and Supplementary Fig. 4k)."

- 1.8 Cytofluorimetric histogram showing the CFSE staining (Figure 2I) is very confusing. From the histogram it is very hard to observe a proper dilution of CFSE dye. How do the authors comment and interpret this data?

Response: Thank you for this question. As T cells proliferate asymmetrically in our system, CFSE histograms usually display poorly resolved generations peaks⁶. For this particular experiment, we analyzed the MFI of

CFSE to avoid arbitrary gating. However, to rule out a possible artifact in our experiments, we repeated this experiment using a BrdU assay to measure proliferation. We obtained the same results as for the CFSE experiment. The data is included into the new Supplementary Fig. 2f.

Changes to the revised manuscript:

- New Supplementary Fig. 2f: BrdU assay of Th9 cells in presence of GW9662 and T0070907.

2. Reviewer

Since their characterization in 2008, Th9 cells have been shown to promote tissue inflammation in both mice and humans. Th9 cells shape disease course in multiple settings, including inflammatory and allergic diseases. While the transcription factor PU.1 has been identified as a major driver for IL-9 production for mouse and human CD4 T cells, most of the in vivo evidence showing the contribution of Th9 cells to disease stems from mouse studies. In this regard, the work presented in this manuscript outlines a very interesting signaling pathway contributing to CD4 T cell-derived IL-9 secretion in human skin inflammation. This study has many strengths:

- 1) It builds on previous mouse studies (notably PMID: 27317260) to extend to a human context the role of glycolysis and mTORC1 signaling in the secretion of IL-9 from Th9 cells.
- 2) It demonstrates the functional role of PPAR-g in driving glycolysis in human Th9 cells.
- 3) It mechanistically uncouples the secretion of IL-9 and IL-13 from effector Th9 cells.
- 4) It ascribes a novel role for IL-9 in enhancing the glycolytic capacity of IL-9R+ T cells

Overall, these findings will further underscore the relevance of the Th9 cell subset in tissue inflammation. The data presented are for the most part convincing and logically build on the previous work from the authors published in Science Immunology.

Response: We thank the reviewer for this positive evaluation and the appreciation of our work.

While the data focusing on Th9 cell biology are straightforward, some sections of the manuscript need to be clarified to better integrate all relevant findings in the field.

My concerns are as follows:

2.1 Some sections of the manuscript are unclear and even conflicting with the message conveyed in the abstract. For instance, to begin their Materials and Methods section, the authors state that their goal was to study human pTH2 cells. Th9 cells are then described as a model to understand “pTh2” cells. Does this mean that the authors regard “pTh2” cells as a distinct CD4 T cell subset (as compared to conventional Th2 cells)? Would this mean that Th9 cells are not a distinct CD4 T cell subset? If the authors believe this, it is then somewhat odd that they rely in the present work on Th9 cells. Defining conditions to generate pTh2 cells in vitro would have been more appropriate.

Response: We thank the reviewer for raising this important question. Seminal work in mice and humans has indeed identified pathogenic T_H2 cells (pT_H2) as a distinct subpopulation within the T_H2 cell subset. As such, they are regarded as distinct from conventional T_H2 (cT_H2) cells^{7,8}. pT_H2 cells differ from their cT_H2 counterparts by their expression of specific cytokines (IL-5, IL-9), cytokine receptors (IL-9R, IL-17RB) and transcription factors (PPAR-γ). As shown in Fig. 1 of this manuscript, these characteristics are shared by T_H9 cells. Given that T_H9 cells also express key T_H2 subset-defining properties⁹ and in particular high levels of PPAR-γ, we propose here to use T_H9 cells as a model for studying the functional role of PPAR-γ in human T_H cell biology.

However, we also observe clear differences between in vitro primed T_H9 cells and pT_H2 cells, such as their lack of IL-33R expression (discussed in the manuscript – line 278-280). These additional differentiation cues remain to be identified.

In summary, we believe that in vitro priming of T_H9 cells recapitulates the key features of the pT_H2 phenotype in type-2 driven diseases. We are grateful to the reviewer for pointing out that some sections in the manuscript, including the Study design in the Methods section, have been confusing for the reader. We have rephrased the suggested paragraphs accordingly (see below).

Changes to the revised manuscript:

- Adapted text, introduction (line 93-94): “Here, we sought to investigate the mechanism by which PPAR-γ regulates the effector function of human T_H9 cells, which share key characteristics with pT_H2 cells.”
- Adapted text, results (line 106-108): “*PPARG*, *IL5*, *IL17RB*, and *IL9R*, which are hallmarks of pT_H2 cells, were upregulated in T_H9 cells as well as in all three pT_H2 datasets, while *IL9* was upregulated in two of the three (Fig. 1c and Supplementary Fig. 1a).”
- Adapted text, results (line 112-113): “In summary, these findings strongly support our hypothesis that in vitro and in vivo primed T_H9 cells share key similarities with pT_H2 cells.”

- Adapted text, discussion (line 267-268): “Here, we used in vivo and in vitro primed T_H9 cells, which represent a subpopulation of PPAR-γ⁺ T_H2 cells and share key characteristics with disease-associated pT_H2 cells, to study the role of PPAR-γ in human T_H cells”.
- Adapted text, methods - study design (line 335-336): “The aim of this study was to investigate the mechanism by which PPAR-γ regulates the effector function of human T_H9 cells, which share key characteristics with pT_H2 cells. We used in vitro and in vivo primed PPAR-γ⁺ T_H9 cells and we performed RNA-seq analysis of activated T_H9 clones upon treatment with the PPAR-γ antagonist GW9662.”

More importantly, while I understand that the authors are focusing on PPARγ, it would be fair to discuss, and experimentally document in Figure 1, the RNA and protein levels of PU.1 considering previous literature (for instance: PMID: 20431622). This would also be important because in their Science Immunology paper, the authors actually reported higher PU.1 expression levels in Th9 cells as compared to Th2 cells at early time points during differentiation.

Response: This is an important remark and we agree that the previous literature on PU.1 requires discussion in our manuscript. In our Science Immunology article⁹ we reported higher *SPI1* mRNA levels (encoding PU.1) at day 3 during T_H9 differentiation as compared to T_H2 primed cells. At day 7, we observed no discernable difference in the expression of *SPI1* levels using RT-qPCR. In the current study, we analyzed in vitro primed T_H1, T_H2, T_H9 and iT_{REG} cells at day 7 by RNAseq. In this data set, *SPI1* is expressed at very low levels in T_H9 cells, albeit slightly but not significantly higher than in T_H2 cells. However, expression levels are very low in T_H2 and T_H9 cells when compared to iT_{REG} cells (Adapted Supplementary Fig. 1a). In addition, *SPI1* is not regularly detected in pT_H2 transcriptomes from human type 2 driven disease. Overall, we conclude that under the conditions used in the present study, *SPI1* is neither specifically expressed by T_H9 cells nor is it associated with the pathogenic phenotype in vivo. As suggested, we discuss these findings in the results section of the manuscript (line 108-110) and show the RNA levels of *SPI1* in Supplementary Fig. 1a.

Changes to the revised manuscript:

- Adapted Supplementary Fig. 1a: *SPI1* expression levels of in vitro primed T_H cell subsets at day 7
- New text, results (line 108-110): “In contrast, SPI1, encoding the transcription factor PU.1, previously shown to be associated with IL-9 expression^{10,11}, was neither expressed in pT_H2-specific transcriptomes, nor in T_H9 cells (Supplementary Fig. 1a).”

2.2 In line with the previous comment, while I understand the authors’ willingness to discuss in detail in the introduction the concept they have written about (reference 2), I think additional balance would be beneficial. Except for my comment on PU.1, I concur that Figure 1 strongly suggests a very close proximity between pTh2 cells and Th9 cells. Surprisingly, despite the presented data, the authors do not discuss this

here while they somehow addressed this in reference 2. Should pTH2 cells be renamed Th9 cells (or the opposite)? Or do the authors think that Th9 cells non-responsiveness to IL-33 is enough to discriminate pTh2 and Th9 cells, especially considering that some laboratories reported that Th9 cells respond to IL-33 (see for instance PMID: 29038366)? Presenting all perspectives would be very important to clarify the field and present a balanced view.

Response: We thank the reviewer for this comment. We now changed the manuscript and highlight that there is a very close proximity between pT_H2 and T_H9 cells and underscore the relevance of T_H9 cells (adapted text, results (line 106-108; 112-113)).

As written in comment 2.1, we observe clear discrepancies between T_H9 and pT_H2 cells, such as the expression of IL-33R, FAR3, and PTGDR2 (reviewed in⁸), suggesting that certain differentiation cues for pTh2 cells are missing in T_H9 cell differentiation. As long as these differentiations are ill defined, we think that it would be hasty to rename either pT_H2 cells or T_H9 cells.

The mentioned discrepancy we observe for the IL-33 responsiveness of T_H9 cells and the previously published study¹² might arise from the different conditions used for in vitro priming. While we used IL-4 and TGF- β to prime T_H9 cells, Ramadan and colleagues additionally added IL-33 during priming to generate IL-33-responsive T_H9 cells.

Changes to the revised manuscript:

- Adapted text, results (line 106-108): “*PPARG*, *IL5*, *IL17RB*, and *IL9R*, which are hallmarks of pT_H2 cells, were upregulated in T_H9 cells as well as in all three pT_H2 datasets, while *IL9* was upregulated in two of the three (Fig. 1c and Supplementary Fig. 1a).”
- Adapted text, results (line 112-113): “In summary, these findings strongly support our hypothesis that in vitro and in vivo primed T_H9 cells share key similarities with pT_H2 cells.”

In summary, this is a very interesting study whose impact could be further enhanced by clearly underscoring the relevance of Th9 cells instead of representing them as a useful tool to study a larger group of inflammatory cells. This is actually what the authors have done in their abstract and graphical abstract, leading to very clear and convincing claims.

3. Reviewer

Bertschi et al. showed the pathological significance of PPAR γ in helper T cell 9 (TH9) cells by performing the functional analysis of PPAR γ using in vitro primed TH9 cells and TH9 cells derived from patients with allergic contact dermatitis.

First, the authors showed that TH9 cells expressed the core feature genes of pathogenic TH2 cells, including enhanced expression of PPAR- γ , IL-5, IL-9, and IL-9R. They showed that glucose uptake in TH9 cells was PPAR- γ dependent. Furthermore, they found that the production of IL-9 and IL-13 by TH9 cells was associated with the glycolytic activity. They also found that glucose and PPAR- γ -dependent production of IL-9 in TH9 cells was regulated via mTORC1. Finally, they showed that IL-9 promotes aerobic glycolysis in IL-9R+ TH cells by inducing the lactate transporter MCT1.

This study demonstrates the importance of the PPAR- γ -mTORC1-IL-9 axis in Th9 cells and is expected to aid in our understanding the type-2-driven skin inflammation. However, additional experiments are needed to support the authors' conclusions. Specific comments are described below.

Response: We thank the reviewer for this positive evaluation and the appreciation of our work.

Major comments:

3.1 The authors addressed the underlying molecular mechanisms by performing experiments of loss-of-function using inhibitors such as GW9662 and MCT1-i. The authors should perform the experiments of gain-of-function, such as the overexpression of PPAR- γ , mTORC1, or MCT1 to confirm the molecular mechanisms that they found.

Response: We thank the reviewer for these important suggestions, which are in line with those made by reviewer 1 in comments 1.1 and 1.2. We therefore kindly refer to our response in comments 1.1 and 1.2.

3.2 In the introduction part, the authors cited previous their-own review paper to introduce pathogenic Th2 cells. But there are other comprehensive review article regarding pathogenic Th2 cells such as Nakayama et al Annu Rev Immunol 2017. The concept of pathogenic Th2 cells was originally proposed by this group, and the paper should be included. Furthermore, in the discussion section, they mentioned the roles of IL-33-IL-33R pathway in both human and mouse pathogenic Th2 cells without the citation of appropriate original article, Endo et al Immunity 2015. They must include appropriate previous work.

Response: Thank you for this valuable comment. We fully agree and have now included the two references in our manuscript (line 66; line 279).

Changes to the revised manuscript:

- New reference in Introduction (line 66): "... referred to as pathogenic T_H2 (pT_H2) cells^{7,8}."
- New reference in Discussion (line 279): "While IL-33R (IL1RL1) is also associated with the pT_H2 phenotype in humans¹³⁻¹⁷..."

3.3 The authors mentioned that mTORC1 activation induces IL-9 expression via HIF-1 α in their Graphical abstract based on previous studies. But they did not present any data related to HIF-1 α in the present study.

Response: Thank you very much for pointing this out. We agree that this could be misleading. We have now changed the graphical abstract accordingly. We have used different color for the HIF-1 α axis and added the reference¹⁸ to make it clear that this pathway was not investigated in the present study, but is based on previous literature. Since the layout of Nature Communications offers no opportunity to separately publish a Graphical Abstract, it is now included in Fig. 6e (adapted text, results (line 261-264)).

Changes to the revised manuscript:

- Adapted Fig. 6e: Graphical abstract, different color for the HIF-1 α axis, including reference
- New text, results (line 261-264): "In addition, we found that PPAR- γ is a positive regulator of aerobic glycolysis in activated human T_H9 cells, which in turn, regulates the expression of IL-9 via mTORC1. Together, this suggests that PPAR- γ and IL-9 facilitate immunometabolic sensing of the tissue microenvironment (Fig. 6e)."

3.4 In Fig. 1G, the expression of IL9 seems to be significantly lower than that of IL5 and IL13. They should investigate the expression of IL9 expression by performing other experiments such as real-time quantitative PCR.

Response: Thank you for this comment. In Fig. 1g, we show the log₂ fold change (FC) of the respective cytokine between the non-lesional control and 24 h, 48 h and 120 h after allergen application, respectively. For *IL5* and *IL13* we therefore see a larger log₂ FC compared with the non-lesional control. However, this analysis does not allow us to draw conclusions about the expression level. We therefore analyzed the counts of *IL9*, *IL5* and *IL13* expression from the RNAseq data and included this data in the manuscript (new Supplementary Fig. 1b).

We see that the *IL9* and *IL5* have similar expression levels, whereas the expression of *IL13* is indeed higher. This is congruent with what we observed in this study (i.e. Fig. 3b) as well as previously⁹. Moreover, we do not claim that *IL9* and *IL5* levels are higher compared to *IL13* levels. Rather, we claim that the expression of *IL9* and *IL5* is higher in T_H9 and pT_H2 cells compared to conventional T_H2 cells.

Changes to the revised manuscript:

- New Supplementary Fig. 1b: Counts of *IL9*, *IL5* and *IL13* expression from RNAseq data

The reviewer also wondered whether the expression of PPAR- γ was upregulated in this experimental setting.

Response: Thank you for raising this very important question. We have not included the *PPARG* data from the skin biopsy studies for two main reasons:

- Most importantly, *PPARG* is expressed in various skin cell types, including keratinocytes that vastly outnumber T_H9 cells in skin punch biopsies as used here (see Reference⁹ and our Immunohistochemistry analysis for PPAR- γ below, left). We believe that the downregulation of *PPARG* we see in our experiment (see below, right) mostly represents the dedifferentiation of keratinocytes in allergic contact dermatitis rather than the expression of *PPARG* in T_H9 cells¹⁹. We included this explanation in our manuscript (line 120-121).
- Secondly, since *PPARG* expression is very tightly controlled and precedes the expression of *IL9*, we believe that an earlier time point than 24 h would have been more appropriate to study *PPARG* levels in T_H9 cells in skin.

Changes to the revised manuscript:

- Adapted text, results (line 120-121): “Due to the fact that *PPARG* is expressed in various skin cell types, including keratinocytes⁹, the correlation analysis of *PPARG* with *IL9* does not allow any conclusion with regard to T_H9 cells.”

3.5 Since the authors have shown that TH9 cells specifically express high level of IL5 in Fig. 1C, they should address the production of IL-5 in Fig. 3a-c and adress it in the manuscript.

Response: Thank you for this comment. We now show the corresponding IL-5 data of Fig.3a-d in the new Supplementary Fig. 3a-d, as well as for Supplementary Fig. 3e and discuss these findings in the results section of the manuscript (line 172-173; 176; 185).

Changes to the revised manuscript:

- New Supplementary Fig. 3a-d: Corresponding IL-5 expression levels of Fig. 3a-d, measured by flow cytometry
- New Supplementary Fig. 3e: IL-5 expression levels of T_H9 cells in presence of LND, measured by flow cytometry
- New text, results (line 172-173): “Similar regulation, albeit less pronounced, was observed for *IL5* expression (Supplementary Fig. 3a-c).”
- Adapted text, results (line 176): “In T_H9 cells primed in vivo, 2-DG and LND inhibited the expression of IL-9 and IL-5 but not IL-13.”
- Adapted text, results (line 185): “Taken together, these observations indicate a dichotomous role of glycolytic activity in regulating the production of IL-9, IL-5 and IL-13 by activated T_H9 cells.”

3.6 In Fig. 3C, the expression of *PPARG* should also be analyzed to investigate the activation of the glycolysis by *PPAR-γ*.

Response: Thank you for this suggestion. We have repeated the experiment and measured the corresponding *PPARG* expression levels, now shown in the new Supplementary Fig. 3c. We further discuss this finding in the result section (line 169-171)

Changes to the revised manuscript:

- New Supplementary Fig. 3c: Corresponding *PPARG* expression levels of Fig. 3c.
- Adapted text, results (line 169-171): “Subsequently, we performed RT-qPCR for *IL9*, *IL13* and *PPARG*. Glucose uptake correlated with the expression of *IL9* and *PPARG*, but not *IL13* (Fig. 3c and Supplementary Fig. 3c).”

3.7 Regarding Fig. 4F and S4D, the authors should explain why they stained for CD3 or CD4 instead of the TH9 cell markers; CCR8 and CCR4 used in Fig. 2H seem to be more useful for markers to detect TH9 cells.

Response: Thank you for raising this point. Indeed, we have shown previously⁹ and in this study that CCR4 and CCR8 are useful markers to identify TH9 cells by flow cytometry. We have been unable to establish specific and sensitive immunofluorescence stainings for CCR8 and CCR4 in ACD. However, on the one hand, we have previously established that virtually all IL-9 producing TH cells in human ACD express PPAR- γ ⁹. On the other hand, new flow cytometry data show that virtually all IL-9⁺ TH cells isolated from ACD skin lesions express pS6, and thus have active mTORC1 signaling (new Supplementary Fig. 4j, line (210-211)). Thus, we have refrained from these immunofluorescence stainings in this study.

Changes to the revised manuscript:

- New Supplementary Fig. 4j: Percentage of pS6⁻ and pS6⁺ cells in CD4⁺IL-9⁺ TH cells isolated from ACD.
- New text, results (line 210-211): "Virtually all IL-9⁺ TH cells isolated from ACD show S6 phosphorylation, and thus have active mTORC1 signaling (Supplementary Fig. 4j)."

3.8 In Fig. 5C, immune-histological sample revealed that many CD3-negative IL-9R-expressing mononuclear cells were accumulated around CD3-positive T cells in the local tissues of allergic contact dermatitis. This result raised the possibility that IL-9 by TH9 cells may affect IL-9 receptor expressing CD3-negative cells, which consequently affect T cell function and glycolytic activity. To address this point, further experiments are required using in vivo primed TH9 clones, to measure the effects of IL-9 signal blockade using neutralizing antibodies against IL-9 or IL-9R blockade on glycolysis and MCT1 expression.

Response: Thank you for this observation and suggestion. The aim of our study is to investigate the regulation of IL-9 by PPAR- γ in TH cells and the effects of paracrine IL-9 on TH cells. With this comment, the reviewer raises the question whether the observed effects of IL-9 on T cell function/glycolysis observed in our study could in fact be mediated via IL-9R⁺ CD3-negative cells indirectly. This question is of course intriguing and worthy of further study.

However, we can exclude that the effects of IL-9 on TH cell function seen in our experiments requires the contribution of CD3-negative cells, as all our experiments (Fig. 5e-m, Fig. 6a-c) were performed in the absence of any CD3-negative cells. We cannot exclude, however, whether in vivo, there is an additional indirect effect via CD3-negative cells.

To investigate the possibility that IL-9R⁺ CD3-negative cells respond to IL-9 and this in turn may affect T cell function, we performed a transwell experiment. IL-9R⁺ T cell clones were co-cultured either in the presence or absence of CD3-negative cells. As an indicator for glycolytic activity, proliferation in response to IL-9 was measured by CFSE dilution. As shown in the data below, we found no difference in the IL-9 response in presence or absence of CD3-negative cells. Rather, we observe slightly lower T cell proliferation when CD3-negative cells are co-cultured and this is compensated by the addition of IL-9. We therefore conclude that the phenotype observed in our study is not induced via CD3-negative cells.

in vivo primed IL-9R⁺ clones

3.9 In Fig. 6D, how many hours after the start of the patch test were the glucose concentrations in the ACD model? The time course was clear in Fig. 1G, but not in this one. Also, were there any steroids or other products applied to the skin before or after the patch test?

Response: The biopsies were taken 48 h post allergen application. We have included this information in the results section (line 255), in the figure legend of Fig. 6d, Supplementary Table 3 as well as in the methods section of the manuscript (line 455). We did not apply steroids or other products to the skin before or during the patch test. This information is now included in the Methods Section of the manuscript (line 455-456).

Changes to the revised manuscript:

- Adapted text, results (line 255): "...the interstitial fluid of tissue homogenates from non-lesional and lesional ACD skin 48 h post allergen application..."
- Adapted text, figure legend of Fig. 6d: "...of interstitial fluids of lesional skin of positive patch test reactions to different allergens (Supplementary Table S3) 48 h post allergen application."
- Adapted Supplementary Table 3: "biopsy collection post allergen application (hours)"

- Adapted text, methods (line 455-456): "...and adjacent non-lesional skin biopsies were taken 48 h after allergen application. No steroids or other products were applied to the skin before or during the patch test."

Minor comments:

- 3.10 In Fig. 1E, the authors performed experiments using in vivo primed Th clones for the first time in this paper. To make it easier for readers to understand, they should add a brief explanation of the experimental system and describe it clearly in the Methods section.

Response: We thank the reviewer for pointing this out. To make it clearer to the reader, we have changed the main text in the result section and included a sentence on how in vivo primed T_H9 clones were generated (line 110-111). We also rewrote the Methods section to explain in more detail how in vivo primed T cell clones were generated (line 362-367). In addition, we reference the gating strategy from our previous paper⁹ in the method section (line 365).

Changes to the revised manuscript:

- New text, results (line 111-112): "...in vivo primed T_H9 clones, which we generated from ex vivo isolated memory T_H cells that were sorted based on chemokine receptor expression (Fig. 1d-f)."
- Adapted text, methods (line 362-367): "Generation of in vivo primed T_H1, T_H17, T_H2 and T_H9 clones: CD4⁺ T cells were isolated from PBMCs using the EasySep™ Human CD4 positive selection kit II (Stemcell Technologies) as per the manufacturer's instructions. Positively selected CD4⁺ T cells were stained for the subsequent sorting of the T_H cell subset. Memory T_H cell subsets were sorted as previously published⁹ with a purity of > 90% according to the expression of chemokine receptors from CD45RA⁻CD25⁻CD8⁻CD3⁺ cells: T_H1 (CXCR3⁺CCR8⁻CCR6⁻CCR4⁻), T_H2 (CXCR3⁻CCR8⁻CCR6⁻CCR4⁺), T_H9 (CXCR3⁻CCR8⁺CCR6⁻CCR4⁺), and T_H17 (CXCR3⁻CCR8⁻CCR6⁺CCR4⁺)."

- 3.11 In Fig. 2C and 2E, the authors showed the maximal glycolytic capacity as the graph. The graph showing the glycolysis rate before addition of Oligomycin should also be presented.

Response: Thank you for this comment. We now included the basal acidification rate of the data shown in Fig 2c (new Supplementary Fig. 2b). In Fig. 2e, however, we did not use oligomycin, but performed an in-run activation experiment with αCD3/CD2/CD28. Therefore, we did not include basal acidification rate for this experiment in the manuscript.

Changes to the revised manuscript:

- New Supplementary Fig. 2b: Basal acidification rate of the data shown in Fig. 2c.

3.12 Fig. S3 seems to be missing.

Response: We named the supplementary figures after the respective main figures. In the initial submitted manuscript, we did not provide supplementary data for Fig. 3. However, in the new revised manuscript, we now show additional data for Fig. 3, which we present in the new Supplementary Fig. 3.

3.13 Regarding Fig. 4 B and C, in lines 175-176, the authors stated "Indeed, IL-9+ T cells were strongly enriched in the pS6+ cell population, whereas the proportion of IL-13+ T cells was similar in the pS6- and pS6+ populations," but in order to make this claim, the proportion of IL-9+ and IL-13+ cells should be shown.

Response: Thank you for pointing this out. In Fig. 4c, we actually display the ratio of IL-9⁺ and IL-13⁺ cells. To make it clearer to the reader, we changed the figure legend and y-axis accordingly.

Changes to the revised manuscript:

- Adapted y-axis and figure legend in Fig. 4c: Ratio (IL-9⁺/ IL-13⁺) highlighted in blue

References:

- 1 LeBrasseur, N. K. *et al.* Thiazolidinediones can rapidly activate AMP-activated protein kinase in mammalian tissues. *Am J Physiol Endocrinol Metab* **291**, E175-181, doi:10.1152/ajpendo.00453.2005 (2006).
- 2 von Meyenn, L., Bertschi, N. L. & Schlapbach, C. Targeting T Cell Metabolism in Inflammatory Skin Disease. *Front Immunol* **10**, 2285, doi:10.3389/fimmu.2019.02285 (2019).
- 3 Angela, M. *et al.* Fatty acid metabolic reprogramming via mTOR-mediated inductions of PPARgamma directs early activation of T cells. *Nat Commun* **7**, 13683, doi:10.1038/ncomms13683 (2016).
- 4 Pan, Y. *et al.* Survival of tissue-resident memory T cells requires exogenous lipid uptake and metabolism. *Nature* **543**, 252-256, doi:10.1038/nature21379 (2017).
- 5 Salas, A. *et al.* JAK-STAT pathway targeting for the treatment of inflammatory bowel disease. *Nat Rev Gastroenterol Hepatol* **17**, 323-337, doi:10.1038/s41575-020-0273-0 (2020).
- 6 Bocharov, G., Luzyanina, T., Cupovic, J. & Ludewig, B. Asymmetry of Cell Division in CFSE-Based Lymphocyte Proliferation Analysis. *Front Immunol* **4**, 264, doi:10.3389/fimmu.2013.00264 (2013).
- 7 Nakayama, T. *et al.* Th2 Cells in Health and Disease. *Annu Rev Immunol* **35**, 53-84, doi:10.1146/annurev-immunol-051116-052350 (2017).
- 8 Bertschi, N. L., Bazzini, C. & Schlapbach, C. The Concept of Pathogenic TH2 Cells: Collegium Internationale Allergologicum Update 2021. *Int Arch Allergy Immunol* **182**, 365-380, doi:10.1159/000515144 (2021).
- 9 Micosse, C. *et al.* Human "TH9" cells are a subpopulation of PPAR-gamma(+) TH2 cells. *Sci Immunol* **4**, doi:10.1126/sciimmunol.aat5943 (2019).
- 10 Chang, H. C. *et al.* PU.1 expression delineates heterogeneity in primary Th2 cells. *Immunity* **22**, 693-703, doi:10.1016/j.immuni.2005.03.016 (2005).
- 11 Chang, H. C. *et al.* The transcription factor PU.1 is required for the development of IL-9-producing T cells and allergic inflammation. *Nat Immunol* **11**, 527-534, doi:10.1038/ni.1867 (2010).
- 12 Ramadan, A. *et al.* Specifically differentiated T cell subset promotes tumor immunity over fatal immunity. *J Exp Med* **214**, 3577-3596, doi:10.1084/jem.20170041 (2017).
- 13 Morgan, D. M. *et al.* Clonally expanded, GPR15-expressing pathogenic effector TH2 cells are associated with eosinophilic esophagitis. *Sci Immunol* **6**, doi:10.1126/sciimmunol.abi5586 (2021).
- 14 Seumois, G. *et al.* Single-cell transcriptomic analysis of allergen-specific T cells in allergy and asthma. *Sci Immunol* **5**, doi:10.1126/sciimmunol.aba6087 (2020).
- 15 Wambre, E. *et al.* A phenotypically and functionally distinct human TH2 cell subpopulation is associated with allergic disorders. *Sci Transl Med* **9**, doi:10.1126/scitranslmed.aam9171 (2017).
- 16 Wen, T. *et al.* Single-cell RNA sequencing identifies inflammatory tissue T cells in eosinophilic esophagitis. *J Clin Invest* **129**, 2014-2028, doi:10.1172/JCI125917 (2019).
- 17 Endo, Y. *et al.* The interleukin-33-p38 kinase axis confers memory T helper 2 cell pathogenicity in the airway. *Immunity* **42**, 294-308, doi:10.1016/j.immuni.2015.01.016 (2015).
- 18 Wang, Y. *et al.* Histone Deacetylase SIRT1 Negatively Regulates the Differentiation of Interleukin-9-Producing CD4(+) T Cells. *Immunity* **44**, 1337-1349, doi:10.1016/j.immuni.2016.05.009 (2016).
- 19 Blunder, S. *et al.* Keratinocyte-derived IL-1beta induces PPARG downregulation and PPARG upregulation in human reconstructed epidermis following barrier impairment. *Exp Dermatol* **30**, 1298-1308, doi:10.1111/exd.14323 (2021).

REVIEWERS' COMMENTS

Reviewer #1 (expertise in immunometabolism, CD4 T cell metabolism):

The authors have attempted to reply to the majority of my additional experiments and requests. The paper has been improved.

Reviewer #2 (expertise in TH9 cells, TH cell subsets):

The authors have carefully addressed my concerns in this revised version of their manuscript. The included data strengthen further the conclusions drawn and this work will stimulate further research in the field.

Reviewer #3 (expertise in allergy, pathogenic TH2 cells):

The revision is satisfactory for this reviewer.

RESPONSE TO REVIEWERS' COMMENTS

Please find below the point-by-point responses to the reviewers' questions.

Changes in the manuscript are marked in blue.

1. Reviewer

The authors have attempted to reply to the majority of my additional experiments and requests. The paper has been improved.

2. Reviewer

The authors have carefully addressed my concerns in this revised version of their manuscript. The included data strengthen further the conclusions drawn and this work will stimulate further research in the field.

3. Reviewer

The revision is satisfactory for this reviewer.